# Single-cell analysis of skeletal muscle macrophages reveals age-associated functional subpopulations

Linda K Krasniewski[1†], Papiya Chakraborty[1†], Chang-Yi Cui[1*], Krystyna Mazan-Mamczarz[1], Christopher Dunn[2], Yulan Piao[1], Jinshui Fan[1], Changyou Shi[1], Tonya Wallace[2], Cuong Nguyen[2], Isabelle A Rathbun[1], Rachel Munk[1], Dimitrios Tsitsipatis[1], Supriyo De[1], Payel Sen[1], Luigi Ferrucci[3], Myriam Gorospe[1*]

[1]Laboratory of Genetics and Genomics, National Institute on Aging Intramural Research Program, National Institutes of Health, Baltimore, United States; [2]Flow Cytometry Core, National Institute on Aging Intramural Research Program, National Institutes of Health, Baltimore, United States; [3]Translational Gerontology Branch, National Institute on Aging Intramural Research Program, National Institutes of Health, Baltimore, United States

*For correspondence:
cuic@grc.nia.nih.gov (C-YC);
myriam-gorospe@nih.gov (MG)

†Co-first authors

Competing interest: The authors declare that no competing interests exist.

**Abstract** Tissue-resident macrophages represent a group of highly responsive innate immune cells that acquire diverse functions by polarizing toward distinct subpopulations. The subpopulations of macrophages that reside in skeletal muscle (SKM) and their changes during aging are poorly characterized. By single-cell transcriptomic analysis with unsupervised clustering, we found 11 distinct macrophage clusters in male mouse SKM with enriched gene expression programs linked to reparative, proinflammatory, phagocytic, proliferative, and senescence-associated functions. Using a complementary classification, membrane markers LYVE1 and MHCII identified four macrophage subgroups: LYVE1−/MHCII^hi (M1-like, classically activated), LYVE1+/MHCII^lo (M2-like, alternatively activated), and two new subgroups, LYVE1+/MHCII^hi and LYVE1−/MHCII^lo. Notably, one new subgroup, LYVE1+/MHCII^hi, had traits of both M2 and M1 macrophages, while the other new subgroup, LYVE1−/MHCII^lo, displayed strong phagocytic capacity. Flow cytometric analysis validated the presence of the four macrophage subgroups in SKM and found that LYVE1− macrophages were more abundant than LYVE1+ macrophages in old SKM. A striking increase in proinflammatory markers (*S100a8* and *S100a9* mRNAs) and senescence-related markers (*Gpnmb* and *Spp1* mRNAs) was evident in macrophage clusters from older mice. In sum, we have identified dynamically polarized SKM macrophages and propose that specific macrophage subpopulations contribute to the proinflammatory and senescent traits of old SKM.

## Editor's evaluation

In this study, Krasniewski and colleagues describe important findings leveraging single-cell transcriptomics to identify subpopulations of macrophages in the skeletal muscle of aging mice. They present solid evidence for the existence of several new resident subpopulations of skeletal muscle macrophages, spanning a range of polarization states using novel markers. Additionally, they identify a shift in relative abundances of these subpopulations with age, leading to a functional shift in inflammatory marker expression and phagocytic capacity. This work will be useful to researchers in the field of immune aging as a resource.

## Introduction

Macrophages are heterogeneous innate immune cells (*Shapouri-Moghaddam et al., 2018*) that provide the first line of defense against pathogens, but are also deeply involved in inflammation, dead cell removal, wound healing, and tissue remodeling (*Mills et al., 2014*; *Ross et al., 2021*; *Shapouri-Moghaddam et al., 2018*). Macrophages adapt to individual tissues and acquire specific tissue-dependent functions (*Wynn et al., 2013*). Upon transplantation, tissue-resident macrophages quickly lose their original gene expression patterns and gain host organ markers (*Lavin et al., 2014*). The tissue environment contributes to determining the tissue-specific protein production by macrophages and thereby establishes tissue-dependent expression patterns and functions (*Gautier et al., 2012*; *Lavin et al., 2014*). Hence, the function of macrophages should be studied in the context of their tissue of residence.

Macrophages play diverse functions in tissues by differentiating into specific functional subgroups, a process usually defined as macrophage polarization (*Yao et al., 2019*). Most macrophages are known to polarize to proinflammatory M1 or anti-inflammatory M2 subgroups (*Martinez et al., 2008*; *Mills et al., 2000*; *Rath et al., 2014*). While such dichotomy largely explains the strikingly different actions of macrophages commonly seen in many tissues, macrophages appear to be more functionally heterogeneous than simply M1 or M2. In this regard, recent flow cytometry and single-cell studies have identified several new macrophage subgroups in arteries, lung interstitium, heart, adipose tissue, and other tissues and organs (*Chakarov et al., 2019*; *Dick et al., 2022*; *Jaitin et al., 2019*; *Lim et al., 2018*; *Schyns et al., 2019*) with distinct tissue-dependent polarization status. Dissecting polarization in each tissue is thus critical to elucidating shared and tissue-specific macrophage functions.

Skeletal muscle (SKM) contains large numbers of macrophages that play critical roles in injury repair and regeneration (*Arnold et al., 2007*; *Tidball, 2011*; *Tidball, 2017*). Macrophages assume different polarization to play distinct functions at different stages of repair after injury (*Scala et al., 2021*; *Yang and Hu, 2018*). In the absence of injury or infection, most macrophages residing in human and mouse SKM were shown to be MRC1 (CD206)+, M2-like macrophages (*Cui et al., 2019*; *Wang et al., 2015*). However, the full range of macrophage subgroups and their age-related changes in SKM is poorly understood (*Cui and Ferrucci, 2020*).

To better understand the complexity of the macrophage polarization status and their changes with aging in mouse SKM, we carried out single-cell transcriptomic analysis. We present evidence that SKM macrophages comprise 11 distinct clusters associated with specific proposed functions. Using a complementary classification based on the presence of membrane markers, SKM macrophages were divided into two large populations based on the presence of LYVE1 and was further classified into four functional subgroups by introducing MHCII as an additional surface marker. We further show that mRNAs that encode proinflammatory proteins and senescence- and aging-related proteins were significantly upregulated in specific macrophage clusters in old SKM. Our findings reveal a dynamic polarization of functional subpopulations of mouse SKM macrophages, including changes toward proinflammatory and senescent phenotypes with aging.

## Results

### Isolation of macrophages from mouse SKM and single-cell RNA sequencing

To isolate macrophages from SKM, we collected all muscles from hind limbs, including quadriceps, gastrocnemius, tibialis, and soleus, from C57BL/6JN male mice, combined and minced them into small cubes, and isolated mononuclear cells by digesting them with enzymes including collagenase and other proteases (*Krasniewski et al., 2022*; *Liu et al., 2015*; *Figure 1A*). To identify macrophage-rich fractions from the mononuclear cell preparation, we carried out flow cytometric analysis based on the presence of CD45, a pan-leukocyte marker, and CD11b, a pan-myeloid lineage marker. As we found previously, CD11b+ cells clearly separated from the rest of the mononuclear cell population (*Krasniewski et al., 2022*).

For single-cell RNA-sequencing (scRNA-seq) analysis, we collected CD11b+ cells from three young (3 months old [3 m.o.]) and three old (23 m.o.) male mice as biological triplicates by fluorescence-activated cell sorting (FACS). From each mouse, 5000–10,000 CD11b+ cells were used for single-cell library preparation using the 3' gene expression pipeline from 10× Genomics followed by RNA-seq

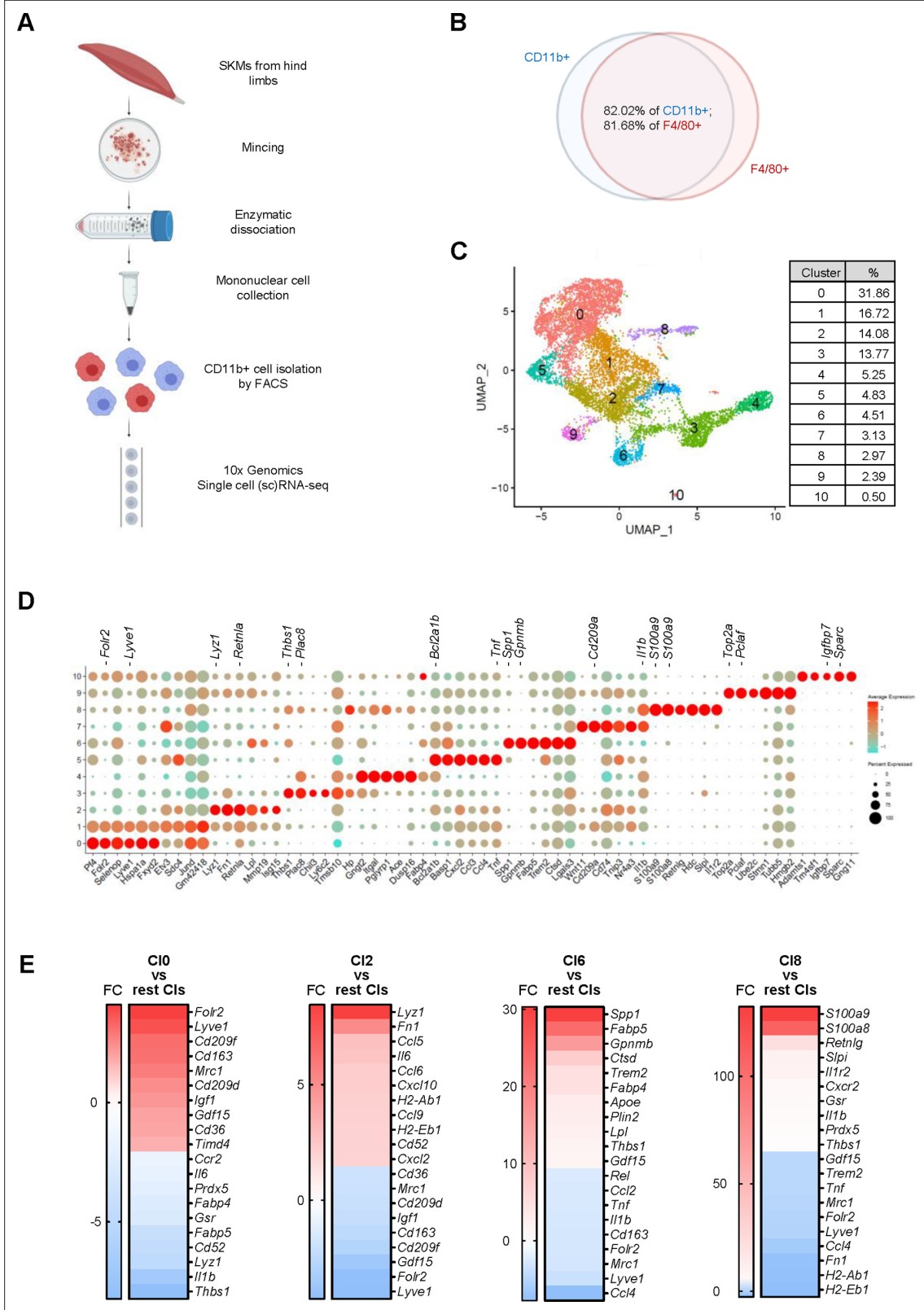

**Figure 1.** Macrophage isolation from mouse skeletal muscle (SKM) and single-cell RNA-seq analysis. (**A**) Workflow of mononuclear cell collection from mouse SKM, CD11b+ cell isolation by FACS, and single-cell RNA-seq analysis using the 10× Genomics platform. (**B**) Cells isolated from mouse SKM that were CD11b+ and F4/80+. (**C**) Unsupervised clustering of SKM macrophages revealed 11 clusters. %, proportion of each cluster. (**D**) Dot plot shows featured mRNAs in each cluster. (**E**) Heat maps show enriched genes in Cl0, 2, 6, and 8.

*Figure 1 continued on next page*

Figure 1 continued

The online version of this article includes the following figure supplement(s) for figure 1:

**Figure supplement 1.** Quality control experiments for the skeletal muscle (SKM) macrophage, single-cell RNA-seq analysis.

**Figure supplement 2.** M2-like features of Cl1 in unsupervised clustering.

analysis. We successfully obtained sequences from 2000 to 5000 single cells from each mouse, and a mean of ~80,000 RNA-seq reads per cell corresponding to a median of >2000 genes per cell (Materials and methods; GEO identifier GSE195507). Sequencing analysis showed that >80% of cells expressing *Cd11b* mRNA were also positive for F4/80 mRNA (*Adgre1* mRNA), another common marker for mouse macrophages (*Figure 1B*). Those cells expressing both *Cd11b* mRNA and *Adgre1* mRNA were considered SKM macrophages. Very few cells were positive for *Ly6g* mRNA or *Siglecf* mRNA (specific markers for neutrophils and eosinophils, respectively; *Figure 1—figure supplement 1A*), indicating minimal contamination from these cells in our macrophage population.

## Identification of 11 macrophage clusters in SKM by unsupervised classification

To gain insight into the subpopulations of SKM macrophages, we pooled scRNA-seq data from young and old mice and performed unsupervised classification. By using FindClusters at a resolution of 0.3, we found 11 clusters (Cl0-10; *Figure 1C*). Given that we isolated macrophages on three different dates due to technical limitations (lengthy procedure) and mouse availability (Materials and methods), we compared the different datasets to ensure there were no batch effects. Overall, the distribution of macrophages across the 11 clusters was comparable among the biological replicates (*Figure 1—figure supplement 1B*), and the patterns of transcriptomes were also comparable among the replicates (*Figure 1—figure supplement 1C*). Those mRNAs that were expressed >1.5-fold higher in a given cluster relative to the other 10 clusters, p<0.05, and were expressed in >25% of macrophages in that cluster are shown in *Supplementary file 1*. Each cluster showed a distinct gene expression pattern (*Figure 1D*).

To investigate the functional features of these clusters, we carried out gene ontology (GO) enrichment analysis using g:Profiler (Materials and methods). Although all clusters shared functional terms general to macrophages, including 'immune system process,' 'defense response,' 'response to stress,' 'cell migration,' and 'cell death,' each cluster also displayed distinct functional associations (*Table 1*). The largest cluster, Cl0, showed a more reparative function, with high expression of M2-type genes (*Mrc1, Cd163, Lyve1*, and *Folr2* mRNAs) and reduced proinflammatory function compared to the other clusters (*Figure 1E* and *Table 1*). The second largest cluster, Cl1, showed a similar expression pattern as Cl0 (*Figure 1D*), including the expression of M2-type mRNAs (*Lyve1* and *Folr2* mRNAs), but the expression levels of these mRNAs were lower in Cl1 than in Cl0 (*Figure 1—figure supplement 2A, B*). This resulted in fewer unique genes in Cl1 when compared to the other 10 clusters (*Supplementary file 1*). When we excluded Cl0 and compared Cl1 with Cl2-10 (*Supplementary file 1*, 'Cl1 vs Cl2-10'), Cl1 showed strong enrichment of M2-type mRNAs (*Figure 1—figure supplement 2C*) and strong association with reparative functions (*Table 1*). Thus, the two largest clusters, Cl0 and Cl1, account for nearly one-half of total macrophages and displayed M2-like gene expression patterns.

Clusters Cl2-9 showed very low expression of M2 marker genes (*Table 1*) and instead displayed more diverse functional associations. Cl2 expressed mRNAs related to inflammation and to the functions of antigen processing and presentation (*Figure 1E* and *Table 1*). The mRNAs present in Cl3 were associated with cellular detoxification, and Cl4 was associated with phagocytosis and expressed elevated MHC class I (MHCI) mRNAs. Cl5 expressed mRNAs strongly associated with the inflammatory response. Cl6 was enriched in mRNAs encoding proteins involved in the response to lipoprotein particles, ATP metabolism, and lipid transport; this cluster also expressed *Gpnmb, Spp1, Ctsd, Trem2*, and *Gdf15* mRNAs, encoding proteins involved in senescence and aging (*Henjum et al., 2016*; *Pazolli et al., 2009*; *Suda et al., 2021*; *Suda et al., 2022*; *Tanaka et al., 2018*; *Williams et al., 2022*), and *Fabp5* and *Fabp4* mRNAs, encoding proteins implicated in atherosclerosis (*Babaev et al., 2011*; *Furuhashi et al., 2007*; *Makowski et al., 2001*; *Figure 1E*). The mRNAs expressed in Cl7 were strongly associated with translation and antigen processing and presentation via MHC class II, while those expressed in Cl8 were associated with cell death and phagocytosis, although M2-type markers

**Table 1.** Gene ontology (GO) annotation of unsupervised clusters.

| Clusters | GO annotation | | Featured membrane proteins |
|---|---|---|---|
| | Elevated | Reduced | |
| Cl0 | Vasculature development (7.8) Amoeboidal-type cell migration (3.7) Endocytosis (3.4) Response to wounding (2.1) | Cytokine production (3.6) Positive regulation of inflammatory response (1.9) Cellular detoxification (1.5) | ↑: LYVE1, FOLR2, MRC1, CD163 ↓: H2-AB1, H2-DMB1 |
| Cl1 | Vasculature development (6.7) Amoeboidal-type cell migration (4.3) Endocytosis (2.9) Response to wounding (1.6) | Translation (4.2) | ↑: LYVE1, FOLR2, MRC1, CD163 |
| Cl2 | Antigen processing and presentation via MHC class II (6.7) Cytokine-mediated signaling pathway (4.3) Cellular response to IL-1 (3.3) Chemotaxis (2.2) Defense response to virus (2.2) | Vasculature development (5.3) Endocytosis (3.4) Muscle cell proliferation (2.5) Amoeboidal-type cell migration (1.4) | ↑: H2-AB1, H2-EB1, H2-DMB1, CCR2 ↓: LYVE1, FOLR2, MRC1, CD163 |
| Cl3 | Cellular detoxification (4.0) Lymphocyte activation (1.6) | Vasculature development (6.9) Muscle cell proliferation (3.9) Response to wounding (1.7) | ↑: CCR2 ↓: LYVE1, FOLR2, MRC1, CD163 |
| Cl4 | Fc receptor signaling pathway (4.2) Regulation of phagocytosis (3.5) Antigen processing and presentation via MHC class I (2.5) Cell killing (1.5) | Angiogenesis (8.6) IL-1 production (5.2) Muscle cell proliferation (2.2) Antigen processing and presentation via MHC class II (2.2) | ↑: H2-K1, H2-D1 ↓: LYVE1, FOLR2, MRC1, CD163, H2-AB1, H2-EB1, H2-DMB1, CCR2 |
| Cl5 | Response to LPS (6.4) TLR signaling pathway (2.9) TNF production (2.0) | Viral entry into host cell (2.3) | ↑: TREM2 ↓: LYVE1, FOLR2, MRC1, CD163 |
| Cl6 | Response to lipoprotein particle (3.3) ATP metabolic process (2.6) Long-chain fatty acid transport (2.0) | Regulation of transcription from RNA polymerase II promoter in response to stress (7.3) Cell chemotaxis (5.1) | ↑: GPNMB, TREM2 ↓: LYVE1, FOLR2, MRC1, CD163, CCR2 |
| Cl7 | Translation (12.4) Antigen processing and presentation via MHC class II (5.2) Ribosome assembly (2.2) | Vasculature development (8.3) IL-1β production (3.9) Response to wounding (2.1) | ↑: H2-AB1, H2-EB1, H2-DMB1 ↓: LYVE1, FOLR2, MRC1, CD163 |
| Cl8 | Positive regulation of cell death (4.2) Phagocytosis (2.2) Autocrine signaling (1.9) | Antigen processing and presentation via MHC class II (4.4) response to IFN-γ (3.3) | ↓: LYVE1, FOLR2, MRC1, CD163, H2-AB1, H2-EB1, H2-DMB1, CCR2 |
| Cl9 | Cell cycle (20.3) DNA replication (4.5) DNA repair (3.2) | Myeloid cell differentiation (3.6) IL-1 production (1.4) | ↓: LYVE1, FOLR2 |
| Cl10 | Vasculature development (19.9) Extracellular matrix organization (9.2) Response to wounding (2.2) | | ↑: LY6C1 |

Parenthesis: negative log10 of adjusted p-value. ↑:elevated in the cluster. ↓:reduced in the cluster.

and MHCII genes were reduced. Of note, *S100a8* and *S100a9* mRNAs, the most robustly elevated mRNAs in Cl8, encode proinflammatory proteins (*Figure 1E*, *Supplementary file 1*). Cl9 expressed cell cycle-related mRNAs, with elevated *Top2a*, *Mki67*, and *Cdk1* mRNAs (*Supplementary file 1*), likely representing a group of reported proliferating macrophages (*Wang et al., 2020*). The smallest

cluster, Cl10 (0.5% of total CD11b+/F4/80+macrophages [*Figure 1C*]) was associated with a reparative function, and one-half of Cl10 cells expressed *Ly6c1* mRNA (*Supplementary file 1*).

Overall, unsupervised clustering revealed a wide functional heterogeneity of SKM macrophages. GO annotation identified clusters of macrophages expressing mRNAs that were particularly associated with reparative functions (Cl0, Cl1, and Cl10), the promotion of inflammation (Cl2 and Cl5), antigen processing and presentation via MHC class II (Cl2 and Cl7), cellular detoxification (Cl3), phagocytosis (Cl4 and Cl8), lipid homeostasis and cell senescence (Cl6), protein synthesis (Cl7), and proliferation (Cl9).

## Identification of M2-like macrophages by membrane marker-based classification

Macrophage membrane markers, including MRC1, CD86, LYVE1, and MHCII, have been successfully used to functionally classify macrophage subgroups (*Mantovani et al., 2002*; *Stein et al., 1992*; *Dick et al., 2022*; *Chakarov et al., 2019*; *Lim et al., 2018*). To complement the unsupervised clustering and gain a more comprehensive view of the highly heterogeneous group of SKM macrophages, we further carried out supervised classification with membrane markers.

Initially, we attempted to subgroup SKM macrophages by traditional polarization markers: MRC1, CD86, or CD80. MRC1 is a widely used marker of M2 macrophages, whereas CD80 and CD86 are M1 markers (*Mantovani et al., 2002*; *Stein et al., 1992*). However, our scRNA-seq data showed that *Mrc1* and *Cd86* mRNAs were broadly expressed in ~80% of macrophages, *Cd80* mRNA was expressed only in a small population, and most macrophages expressed *Mrc1* and *Cd86* mRNAs simultaneously (*Figure 2—figure supplement 1A*), suggesting they are not ideal to classify SKM macrophages at the transcriptomic level.

We therefore turned to other candidate membrane markers. Recently, LYVE1 and MHCII were successfully used to subgroup several tissue-resident macrophages (*Dick et al., 2022*; *Chakarov et al., 2019*; *Lim et al., 2018*). By unsupervised clustering, *Lyve1* and MHCII mRNAs were differentially expressed in select clusters (*Table 1*); therefore, we classified SKM macrophages by LYVE1 expression levels first. LYVE1 status divided SKM macrophages into two large, similarly sized groups, LYVE1+ (46.6%) and LYVE1− (53.4%) (*Figure 2A*). LYVE1+ macrophages displayed an M2-like transcriptomic program, including mRNAs encoding proteins associated with functions in 'vasculature development,' 'wound repair,' and 'endocytosis' (*Figure 2B*; *Buchacher et al., 2015*; *Stein et al., 1992*). Interestingly, transcripts encoding proangiogenic proteins (*Ang*, *Stab1*, and *Egr1* mRNAs) as well as transcripts encoding antiangiogenic proteins (*Cfh* and *Hspb1* mRNAs) were upregulated in LYVE1+ macrophages. Transcripts encoding proteins implicated in wound healing (*Igf1*, *Nrp1*, and *Gas6* mRNAs) were also elevated in LYVE1+ macrophages (*Figure 2B*); and mRNAs encoding endocytosis-related members of the CD209 family (including *Cd209d* and *Cd209b* mRNAs) as well as *Cd36*, *Cd163*, and *Mrc1* mRNAs were also highly expressed in LYVE1+ macrophages. Other mRNAs, such as *Timd4* and *Fcna* mRNAs, were almost exclusively expressed in the LYVE1+ macrophages and might be good candidate markers for this population (*Figure 2—figure supplement 1B*).

In contrast, LYVE1− macrophages expressed higher levels of mRNAs encoding antigen-processing and antigen-presenting proteins (*H2-Eb1*, *H2-Ab1*, *H2-DMb1*, and *Cd74* mRNAs), proteins related to the NF-kB signaling pathway and implicated in cell death (*Il1b*, *Bcl2a1b*, *Bcl2a1d*, *Cd14*, *Traf1*, and *Malt1* mRNAs), and proteins with function in antioxidant responses (*Gsr*, *Prdx5*, *Prdx6*, *Prdx1*, and *Hp* mRNAs). In addition, many mRNAs encoding ribosomal proteins were highly expressed in this group (*Figure 2—figure supplement 1C*). The full list of mRNAs differentially abundant in LYVE1+ and LYVE1− macrophages is in *Supplementary file 2*.

To validate the differences in gene expression programs, we separated each population (LYVE1+ and LYVE1− macrophages) by FACS. Reverse transcription (RT) followed by real-time quantitative (q)PCR analysis confirmed that *Lyve1*, *Folr2*, *Timd4*, *CD209f*, and *Fcna* mRNAs were almost exclusively expressed in LYVE1 + macrophages (*Figure 2D*, top, n=2 biological replicates). By contrast, *Mrc1*, *Igf1*, and *Ang* mRNAs were expressed in both LYVE1+ and LYVE1− macrophages, but at much higher levels in LYVE1+, while *Il1b* mRNA levels were significantly higher in the LYVE1− population (*Figure 2D*, bottom). The RT-qPCR results (*Figure 2D*) were consistent with the single-cell transcriptomic analysis (*Figure 2B and C*; *Supplementary file 2*), indicating that LYVE1 is an effective marker for subgrouping mouse SKM macrophages.

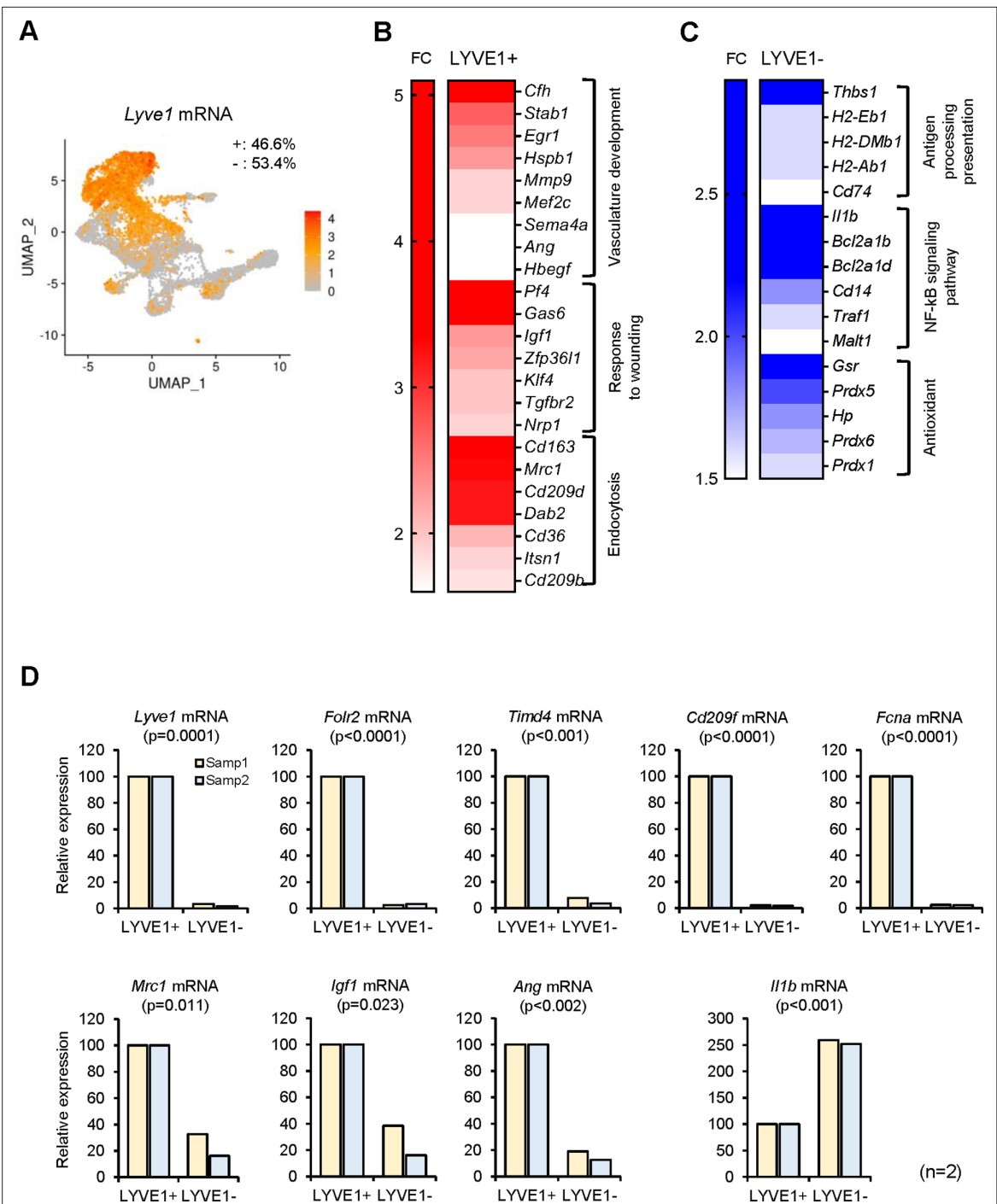

**Figure 2.** Functional clusters of genes differentially expressed in LYVE1+ and LYVE1− macrophages following single-cell RNA-sequencing (scRNA-seq) analysis. (**A**) *Lyve1* mRNA expression pattern in skeletal muscle (SKM) macrophages. (**B**) mRNAs highly expressed in functional clusters of LYVE1+ macrophages. (**C**) mRNAs highly expressed in LYVE1− macrophages. (**D**) Validation of select mRNAs differentially abundant as identified in panels (**B and C**). LYVE1+ and LYVE1− macrophages were isolated by fluorescence-activated cell sorting (FACS) from three male mice, 3 months old (m.o.), and mRNAs elevated in LYVE1+ macrophages (top and bottom left), and mRNAs predominantly elevated in LYVE1− macrophages (bottom right) were quantified by RT-quantitative PCR (qPCR) analysis. Data were normalized to the levels of *Gapdh* mRNA, also measured by RT-qPCR analysis. Data represent the means and SD from two different sorts for each group.

The online version of this article includes the following figure supplement(s) for figure 2:

**Figure supplement 1.** mRNAs highly expressed in LYVE1+ or LYVE1− macrophages.

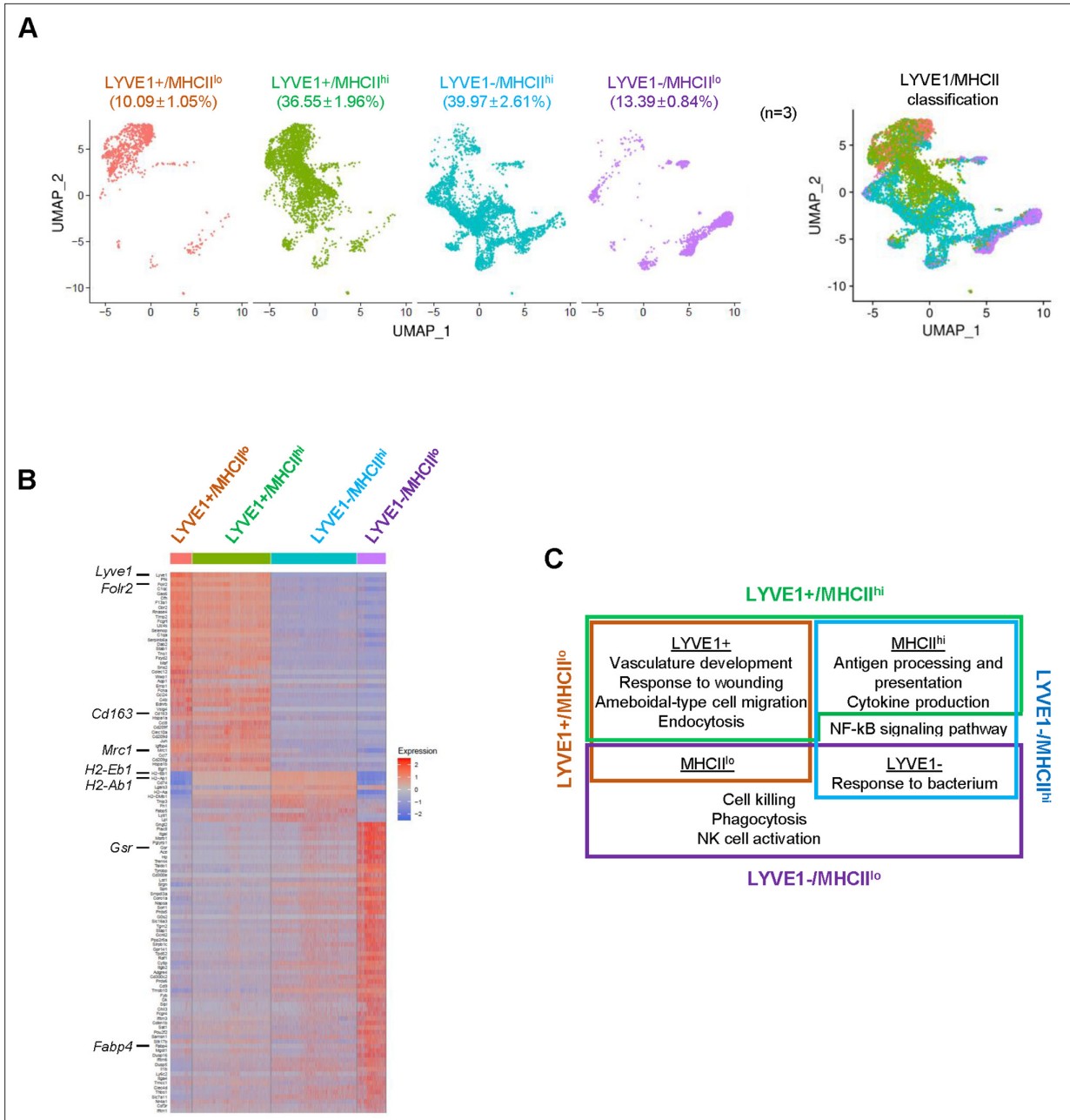

**Figure 3.** Classification of mouse skeletal muscle (SKM) macrophages into four functional subgroups according to surface markers. (**A**) Subclassification of mouse SKM macrophages based on LYVE1 and MHCII levels: LYVE1+/MHCII<sup>lo</sup>, LYVE1+/MHCII<sup>hi</sup>, LYVE1−/MHCII<sup>hi</sup>, and LYVE1−/MHCII<sup>lo</sup>. Uniform Manifold Approximation and Projection (UMAP) analysis of the distribution and size of each of four subgroups individually (left) and combined (right). (**B**) Heat map analysis of the single-cell RNA-sequencing (scRNA-seq) data depicting distinct gene expression patterns of the four subgroups. (**C**) Gene ontology (GO) annotation of the functions of each subgroup. Brown box, LYVE1+/MHCII<sup>lo</sup>; green box, LYVE1+/MHCII<sup>hi</sup>; blue box, LYVE1−/MHCII<sup>hi</sup>; purple box, LYVE1−/MHCII<sup>lo</sup>.

The online version of this article includes the following figure supplement(s) for figure 3:

**Figure supplement 1.** LYVE1 and MHCII are used to classify skeletal muscle (SKM) macrophages into four subgroups.

## LYVE1 and MHCII further classify macrophages into four subgroups

By single-cell profiling analysis, the membrane marker MHCII (encoded by *H2-Ab1* and *H2-Eb1* mRNAs) divided SKM macrophages into two groups, MHCII<sup>hi</sup> and MHCII<sup>lo</sup> (*Figure 3—figure supplement 1A*). Considering the relative levels of LYVE1 and MHCII on the membrane allowed the classification of SKM macrophages into four subgroups: LYVE1+/MHCII<sup>hi</sup>, LYVE1+/MHCII<sup>lo</sup>, LYVE1−/MHCII<sup>hi</sup>, and LYVE1−/

MHCII$^{lo}$ (*Figure 3A*). Among them, LYVE1+/MHCII$^{hi}$ and LYVE1−/MHCII$^{hi}$ were the largest subgroups, comprising 36.55 and 39.97% of all macrophages, respectively (*Figure 3A*), while LYVE1+/MHCII$^{lo}$ and LYVE1−/MHCII$^{lo}$ comprised 10.09 and 13.39%, respectively (biological replicates in *Figure 3—figure supplement 1B, C*). The overall distribution of cells among the biological replicates was comparable among the subgroups selected based on expression levels of *Lyve1* and *MhcII* mRNAs in supervised analysis (*Figure 3—figure supplement 1B, C*). Notably, both LYVE1+ subgroups, LYVE1+/MHCII$^{lo}$ and LYVE1+/MHCII$^{hi}$, largely overlapped with two reparative clusters, Cl0 and Cl1, from the unsupervised clustering (compare *Figure 3A* with *Figure 1C*). The LYVE1−/MHCII$^{hi}$ subgroup comprised most macrophages from Cl2, Cl5, Cl6, Cl7, Cl9, and part of Cl3. LYVE1−/MHCII$^{lo}$ contained Cl4 and part of Cl3 and Cl8. Overall, LYVE1− macrophages showed more heterogeneity than LYVE1+ macrophages (*Figures 1C and 3A*).

Single-cell analysis (*Figure 3B*) revealed distinct gene expression patterns across the four supervised subgroups. Those mRNAs that were expressed >1.5-fold higher in a given subgroup relative to the other three subgroups (p<0.01) and were expressed in >25% of macrophages in that subgroup are shown in *Supplementary file 3*. Functional annotations of the genes showing higher expression in each subgroup revealed that LYVE1+/MHCII$^{lo}$ macrophages (brown box, *Figure 3C*) expressed higher levels of mRNAs associated with vasculature development and wound healing, similar to the macrophages in Cl0 and Cl1 (*Table 1*) and M2 macrophages (*Krzyszczyk et al., 2018*). LYVE1−/MHCII$^{hi}$ macrophages (blue box, *Figure 3C*) were associated with antigen processing and presentation, cytokine production, and responses to bacteria and were overall more M1-like (*Mills, 2015*). LYVE1+/MHCII$^{hi}$ macrophages (green box, *Figure 3C*) were a more complex group; GO annotation suggested that they largely shared LYVE1+/MHCII$^{lo}$ (M2-like) functions like vasculature development and wound healing, but also shared LYVE1−/MHCII$^{hi}$ (M1-like) functions such as antigen processing and presentation and cytokine production. Finally, LYVE1−/MHCII$^{lo}$ macrophages (purple box, *Figure 3C*) were associated with cytotoxicity and phagocytosis. Notably, among the four subgroups, LYVE1+/MHCII$^{hi}$ and LYVE1−/MHCII$^{lo}$ were not previously reported in SKM (*Wang et al., 2020*), and LYVE1−/MHCII$^{lo}$ macrophages were not reported in any other tissue so far (*Chakarov et al., 2019*; *Lim et al., 2018*). Thus, in addition to the M2-like (LYVE1+/MHCII$^{lo}$) and M1-like (LYVE1−/MHCII$^{hi}$) subgroups, supervised classification revealed two new subgroups, LYVE1+/MHCII$^{hi}$ and LYVE1−/MHCII$^{lo}$, in resting mouse SKM. The supervised classification thus complemented the unsupervised clustering, offering a more comprehensive understanding of the heterogeneity of SKM macrophages.

## Confirmation of four SKM macrophage subgroups by flow cytometry

We further analyzed if the macrophage subgroups identified from scRNA-seq could be validated by cell-surface protein markers. As anticipated, flow cytometric analysis using antibodies that recognized LYVE1 and MHCII divided CD45+/CD11b+/F4/80+SKM macrophages from 3 m.o. male mice into four subgroups, LYVE1+/MHCII$^{lo}$, LYVE1+/MHCII$^{hi}$, LYVE1−/MHCII$^{hi}$, and LYVE1−/MHCII$^{lo}$ (*Figure 4A*, n=4). Notably, the LYVE1+/MHCII$^{lo}$, LYVE1−/MHCII$^{hi}$, and LYVE1−/MHCII$^{lo}$ subgroups showed clear clusters of cells, but LYVE1+/MHCII$^{hi}$ macrophages spread across LYVE1+/MHCII$^{lo}$ and LYVE1−/MHCII$^{hi}$ (*Figure 4A*, bottom). The sizes of each subgroup identified by flow cytometry and those identified by single-cell transcriptomics were comparable (*Figures 3A and 4A*). While the present study focused on SKM macrophages from male mice, we assessed the overall influence of sex on macrophage polarization in SKM by performing flow cytometric analysis with SKM macrophages from 3 m.o. female mice. As shown, female mice also showed four SKM macrophage subgroups, comparable to male mice (compare *Figure 4A* with *Figure 4—figure supplement 1A*; n=4). However, when compared with male SKM macrophages, female SKM LYVE1+/MHCII$^{hi}$ macrophages were ~17% lower, and LYVE1−/MHCII$^{lo}$ macrophages were ~42% higher (*Figure 4—figure supplement 1B*). It was recently reported that mouse gender affects macrophage polarization, function, and morphology (*Han et al., 2021*; *Jaillon et al., 2019*). The biological significance of the sex-related differences in specific macrophage subgroups in SKM warrants further study.

To validate these macrophage subgroups in intact mouse SKM, we performed immunofluorescence detection analysis (*Figure 4B*). As anticipated, both LYVE1+ cells (red) and MHCII+ cells (green) were found in intramuscular connective tissues, namely the endomysium and perimysium regions, as visualized with discontinuous lines demarking muscle cell membranes (*Figure 4B*). Importantly, many LYVE1+ cells were also MHCII+ (LYVE1+/MHCII$^{hi}$) in SKM (*Figure 4B*, yellow arrows, top), consistent

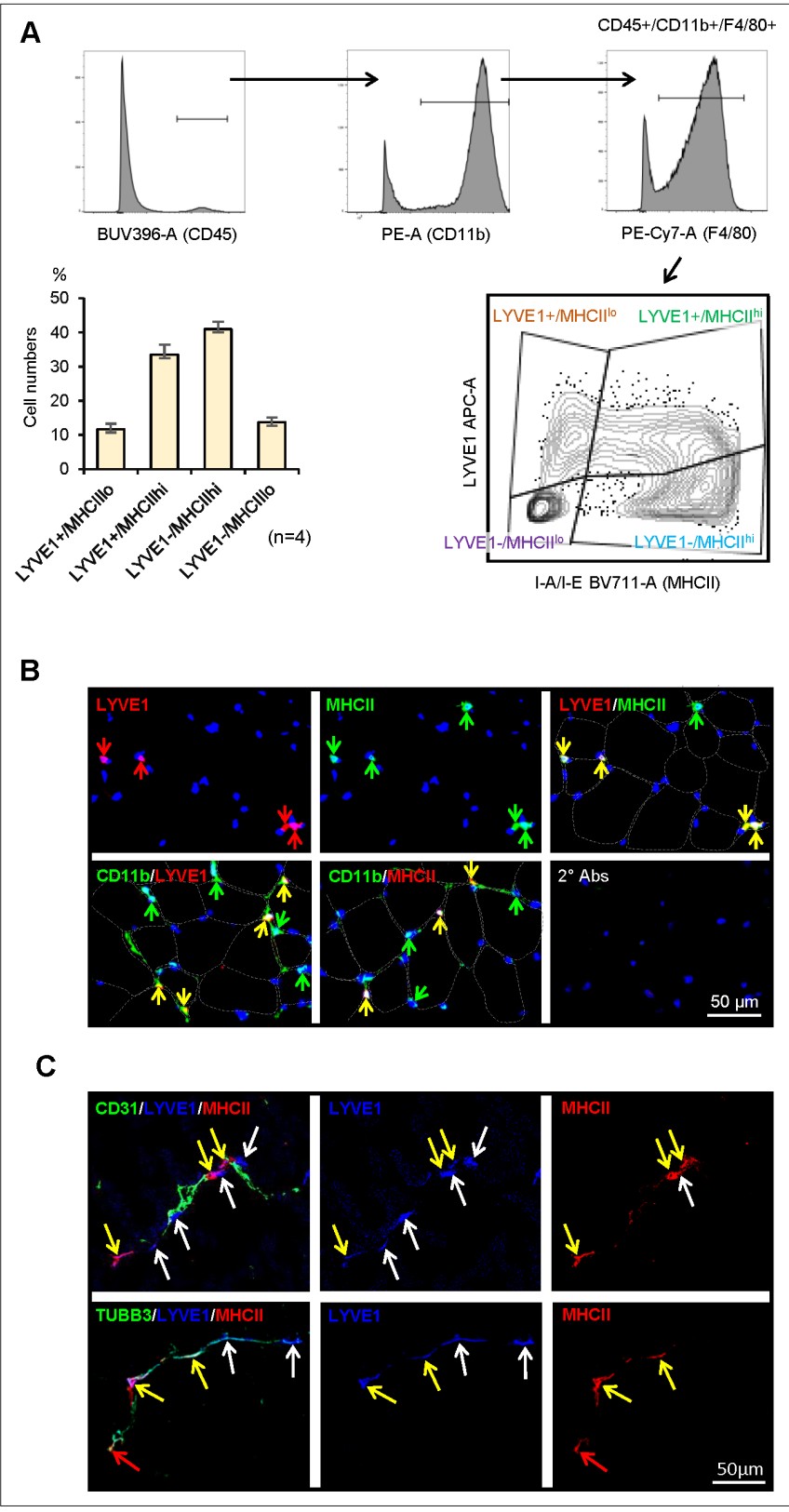

**Figure 4.** Characterization of macrophage subgroups by flow cytometry and immunofluorescence staining.
(**A**) Flow cytometric analysis of the four subgroups in skeletal muscle (SKM). CD45+/CD11b+/F4/80+macrophages
(top three panels show gating) were further classified by LYVE1 and MHCII (bottom right). LYVE1+/MHCII^lo,
LYVE1−/MHCII^hi, and LYVE1−/MHCII^lo subgroups formed clear cell clusters, while LYVE1+/MHCII^hi spanned

*Figure 4 continued on next page*

*Figure 4 continued*

LYVE1+/MHCII$^{lo}$ and LYVE1−/MHCII$^{hi}$. Note: the sizes of each subgroup by flow cytometric analysis (bottom left) were similar to those seen with single-cell RNA-seq analysis. Gating was based on FMO (fluorescence minus one) controls for each experiment. (**B**) Immunofluorescence analysis of the presence of LYVE1+/MHCII$^{hi}$ macrophages in mouse SKM. Top, LYVE1+, MHCII+, and LYVE1+/MHCII+ double-positive cells in endomysium and perimysium areas of mouse SKM. Bottom, colocalization of LYVE1 (left) and MHCII (middle) with CD11b, a macrophage marker; secondary antibodies only (right). (**C**) LYVE1+ macrophages LYVE1+/MHCII$^{lo}$ and LYVE1+/MHCII$^{hi}$, colocalizing with CD31+, depicting blood vessels (top). LYVE1+ and LYVE1− macrophages colocalizing with the nerve fiber marker TUBB3+ (bottom).

The online version of this article includes the following figure supplement(s) for figure 4:

**Figure supplement 1.** Skeletal muscle (SKM) macrophages from female mice.

with the flow cytometric and the single-cell transcriptomic analyses. Tyramide signal amplification (TSA) staining of CD11b confirmed that LYVE1+ and MHCII + cells were CD11b+ (***Figure 4B***, bottom). Thus, immunofluorescence analysis indicated that LYVE1+, MHCII+, and LYVE1+/MHCII$^{hi}$ macrophages were constitutively present in mouse SKM. Further analysis revealed that both LYVE1+/MHCII$^{lo}$ (white arrows) and LYVE1+/MHCII$^{hi}$ (yellow arrows), but not LYVE1−/MHCII$^{hi}$ macrophages, localized near CD31+ blood vessels (***Figure 4C***, top). However, both LYVE1+ (yellow and white arrows) and LYVE1− (red arrows) macrophages localized near nerve fibers, the latter positive for TUBB3 (***Figure 4C***).

## Macrophage subgroups show distinct phagocytic capacities

To gain insight into the functional differences among the four subgroups, we assessed their phagocytic capacity, a fundamental function of macrophages, using a flow cytometry-based method that measures the uptake of labeled particles (pHrodo Red *Escherichia coli* Bioparticle assay, Materials and methods). As anticipated, all macrophage subgroups were strongly phagocytic (***Figure 5A***), with 97.2% of LYVE1+/MHCII$^{lo}$, 98.5% of LYVE1+/MHCII$^{hi}$, 86.4% of LYVE1−/MHCII$^{hi}$, and 49.6% of LYVE1−/MHCII$^{lo}$ macrophages actively phagocytizing *E. coli* particles at 37°C; in control incubations, <17.7% macrophages were active at 4°C (***Figure 5A and B***, n=3). Significantly, fewer macrophages in the LYVE1−/MHCII$^{lo}$ subgroup were actively phagocytic compared with the other three subgroups (***Figure 5B***, p<0.01), but those macrophages that were active showed greater phagocytic capacity than the other three subgroups.

As macrophages showed a range of phagocytic capacities, we divided them into four groups by their geometric mean fluorescence intensity (gMFI): negative (Neg; intensity <10$^3$), low (Lo; 10$^3$–10$^4$), medium (Med; 10$^4$–10$^5$), and high (Hi;>10$^5$; ***Figure 5A***). The phagocytic capacities of the four macrophage subgroups were similar (***Figure 5C***, n=3), and LYVE1+/MHCII$^{lo}$, LYVE1+/MHCII$^{hi}$, and LYVE1−/MHCII$^{hi}$ subgroups showed similar numbers of active macrophages in each of the low-, medium-, and high-capacity groups (***Figure 5D***). However, LYVE1−/MHCII$^{lo}$ macrophages showed significantly fewer active macrophages in the Lo group and strikingly more in the Hi capacity group compared to the other three subgroups (***Figure 5D***). This finding suggested that the LYVE1−/MHCII$^{lo}$ group comprised two macrophage subpopulations with different phagocytic capacity: a silent group and a highly phagocytic group, each with roughly the same number of macrophages (***Figure 5A***, top). We performed efferocytosis assays to further assess the capacity of the macrophage subgroups in phagocytizing apoptotic cells. All four macrophage subgroups showed lower efferocytosis than phagocytosis, but LYVE1−/MHCII$^{lo}$ macrophages again showed relatively greater capacity (***Figure 5—figure supplement 1A, B***).

These observations prompted us to further subclassify the LYVE1−/MHCII$^{lo}$ subgroup by unsupervised clustering, which yielded six subclusters (SubCl; ***Figure 5—figure supplement 2A***). GO annotation showed clustering of phagocytosis-related terms only in SubCl0 (***Figure 5—figure supplement 2B***). GO annotation suggests that SubCl0 may represent macrophages with higher phagocytic capacity in the LYVE1−/MHCII$^{lo}$ subgroup (***Figure 5A, B and D***), although further studies are required for clarification.

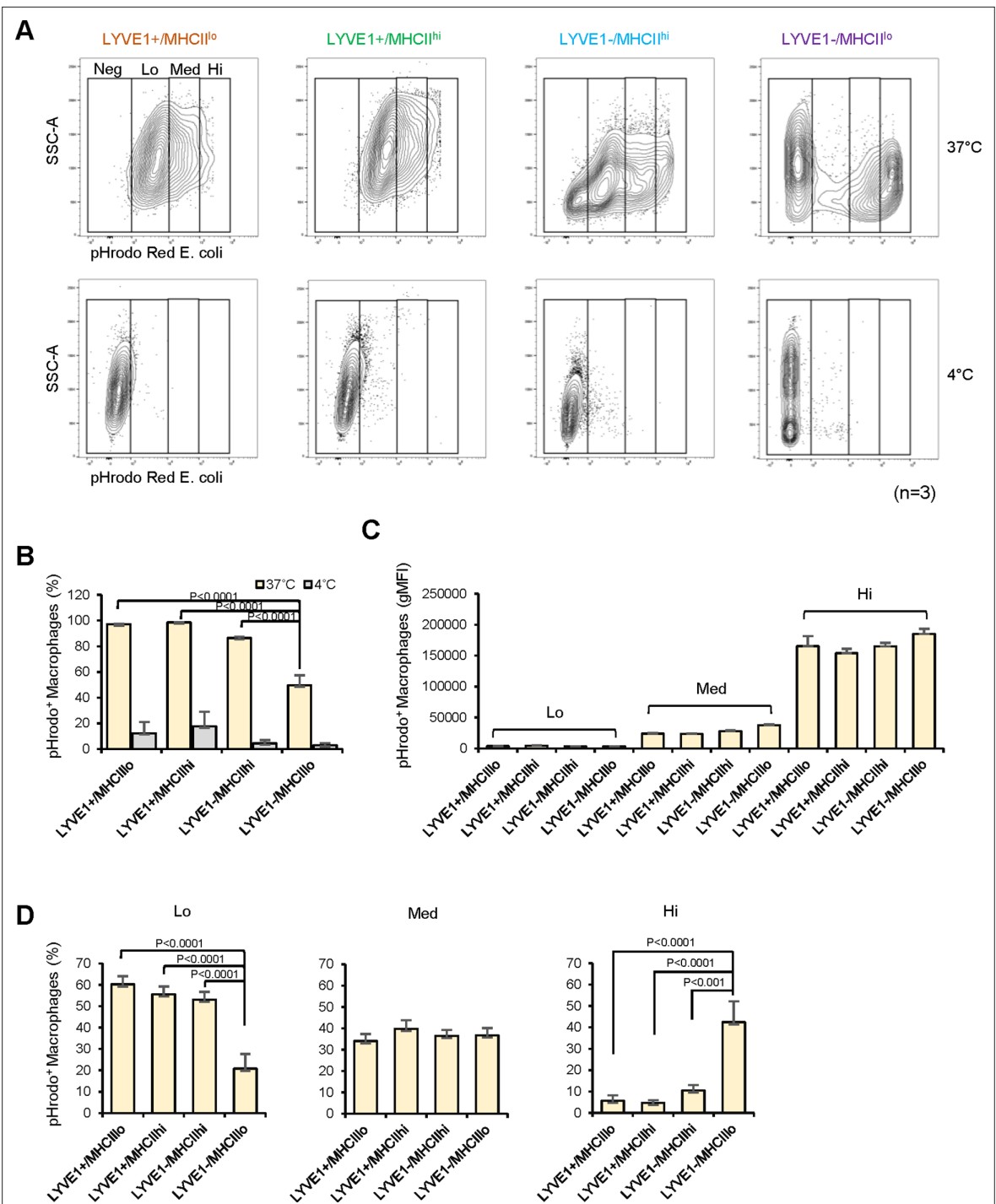

**Figure 5.** Analysis of the phagocytic capacities of each macrophage subgroup. (**A**) Phagocytic activity was measured for mouse skeletal muscle (SKM) macrophages at 4°C (control, low phagocytosis) and 37°C (active phagocytosis, right boxes). Phagocytic capacity was divided into groups that were negative (Neg; intensity <$10^3$), low (Lo; $10^3$–$10^4$), medium (Med; $10^4$–$10^5$), and high (Hi; >$10^5$), depending on signal intensities. Gating was established using fluorescence minus one (FMO) controls for each experiment. (**B**) Quantification of the macrophages showing active phagocytosis (Lo + Med + Hi) in the four subgroups. (**C**) Signal intensities of macrophages in each capacity group (Lo, Med, and Hi). (**D**) Quantification of number of active phagocytic macrophages in each subgroup of the three intensity groups. Data are representative of three independent experiments.

The online version of this article includes the following figure supplement(s) for figure 5:

**Figure supplement 1.** Efferocytotic capacities of four macrophage subgroups.

**Figure supplement 2.** Classification of LYVE1−/MHCII$^{lo}$ subgroup by unsupervised clustering.

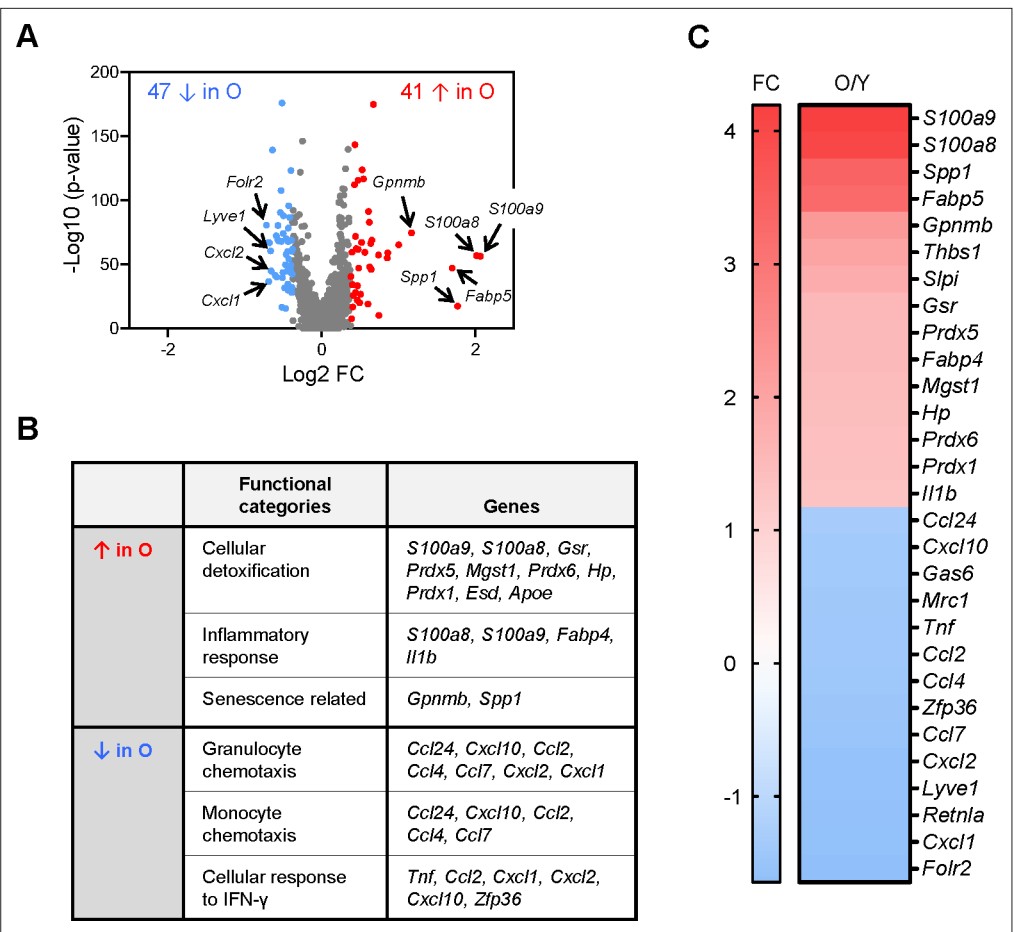

**Figure 6.** Analysis of gene expression programs in skeletal muscle (SKM) macrophages from young and old mice before clustering. (**A**) In single-cell RNA-sequencing (scRNA-seq) analysis, a total of 88 mRNAs were differentially expressed between old and young SKM. Arrows indicate featured mRNAs upregulated (red) or downregulated (blue) in old SKM macrophages. (**B**) Gene ontology (GO) annotation depicting the functional categories that were upregulated and downregulated in the old SKM macrophages relative to young SKM macrophages. (**C**) Fold changes in the abundance of select mRNAs (O/Y), as determined from the scRNA-seq analysis.

The online version of this article includes the following figure supplement(s) for figure 6:

**Figure supplement 1.** Number of CD45+, CD45+/CD11b+, and CD45+/CD11b+/F4/80+ cells obtained from young and old males and females.

## Elevated proinflammatory and senescence-related mRNAs in old SKM macrophages

To investigate if there are aging-related changes in SKM macrophages, we further analyzed the genes differentially expressed in macrophages from young and old mouse SKM. The number of live macrophages isolated from SKM was comparable between young and old mice, both in males and females (*Figure 6—figure supplement 1A-D*, n=5), and the number of differentially abundant mRNAs was rather small, likely reflecting the lower sensitivity of scRNA-seq analysis. Therefore, we used slightly less strict criteria to find differentially expressed mRNAs: those expressed in >10% of total macrophages in young or old, p<0.01, and fold change >1.3. By these criteria, 41 mRNAs were more abundant, and 47 mRNAs were less abundant in macrophages from old SKM (*Figure 6A*). GO annotation suggested that mRNAs encoding proteins involved in chemotaxis of granulocytes (e.g. *Cxcl1* and *Cxcl2* mRNAs; *Girbl et al., 2018*) and monocytes (e.g. *Ccl2* and *Ccl7* mRNAs; *Deshmane et al., 2009*), and the cellular response to IFN-γ (e.g. *Tnf*, *Cxcl10*, and *Zfp36* mRNAs) were less abundant in old SKM macrophages (*Figure 6B and C*). Some mRNAs encoding M2-like markers (e.g. *Lyve1*, *Folr2*, and *Mrc1* mRNAs) were also significantly lower in old SKM macrophages (*Figure 6A and C*). By

contrast, mRNAs encoding proteins related to cellular detoxification (e.g. *Gsr*, *Hp*, *Prdx1*, *Prdx5*, and *Prdx6* mRNAs), inflammation (e.g. *S100a8*, *S100a9*, *Fabp4*, and *Il1b* mRNAs), senescence (*Gpnmb* and *Spp1* mRNAs), and long-chain fatty acid transporters (*Fabp4* and *Fabp5* mRNAs; *Wang et al., 2018*; *Babaev et al., 2011*; *Furuhashi et al., 2007*; *Pazolli et al., 2009*; *Suda et al., 2021*; *Suda et al., 2022*) were elevated in old SKM macrophages (*Figure 6B and C*; full list in *Supplementary file 4*).

We next analyzed the relative abundance of macrophage subgroups as a function of age. scRNA-seq indicated that LYVE1+ macrophages decreased, while LYVE1− macrophages increased in old SKM (*Figure 7A*). Flow cytometric analysis confirmed this trend, as LYVE1+ macrophages decreased and LYVE1− macrophages increased in old SKM (*Figure 7B* —supporting *Figure 1A and B*, n=4). Thus, both scRNA-seq and flow cytometric analysis confirmed the changes in numbers of LYVE1+ and LYVE1− macrophages in old SKM, consistent with the changes in *Lyve1*, *Folr2*, and *Mrc1* mRNAs during aging (*Figures 6A, C, 7A and B*). All four macrophage subgroups displayed differentially expressed mRNAs. The top 15 elevated and top 10 reduced mRNAs in each subgroup were shown (*Figure 7—figure supplement 1C*). *S100a9* mRNA, encoding a proinflammatory marker, was upregulated in all four subgroups, the senescence-related *Gpnmb* and *Spp1* mRNAs and the fatty acid transporter *Fabp5* mRNA were elevated in two MHCII^hi subgroups, LYVE1+/MHCII^hi and LYVE1−/MHCII^hi, while *Apoe* and *Fabp4* mRNAs were only abundant in LYVE1−/MHCII^hi macrophages, and *Il1b* mRNA was elevated only in LYVE1−/MHCII^lo macrophages in old SKM (*Figure 7—figure supplement 1C*).

In unsupervised clustering, Cl0 macrophages, mostly *Lyve1+/Folr2+/Mrc1+*, were less abundant in old SKM, while macrophages in Cl3, 6, and 8 increased in old SKM (*Figure 7C*). *Gpnmb*, *Spp1*, and *Fabp5* mRNAs were largely concentrated in Cl6, a cluster that was strikingly enriched in old SKM (*Figure 7D*), and *S100a9* and *S100a8* mRNAs were elevated mainly in Cl8 in old SKM (*Figure 7D*). Biological replicates of the expression patterns of these genes in young and old SKM (Cl6 and Cl8) are shown (*Figure 7—figure supplement 2A,B*, respectively).

Overall, gene expression changes suggest that mRNAs related to chemotaxis and responses to pathogens were reduced, but mRNAs encoding proinflammatory, senescence, and cellular detoxification were elevated in macrophages from old SKM. In old SKM macrophages, senescence-related mRNAs were enriched in Cl6 and proinflammatory mRNAs in Cl8.

## Discussion

Heterogeneity and functional versatility are critical characteristics of macrophages. Derived from embryonic and/or adult hematopoietic system (*Cox et al., 2021*), macrophages adapt their gene expression profiles to the tissues in which they reside and play diverse functions by polarizing to different subgroups. In this study, we identified functional subgroups of mouse SKM macrophages by single-cell transcriptomic analysis. Using unbiased clustering, we found 11 clusters, each comprising macrophages associated with reparative, proinflammatory, phagocytic, proliferative, and lipid homeostasis and senescence/aging functions, revealing the striking heterogeneity of SKM macrophages. An alternative classification based on membrane markers further revealed populations that expressed or lacked LYVE1 on their plasma membrane and could be further divided into four subgroups by the levels of cell-surface MHCII proteins. These four subgroups included the well-known M2-like and M1-like macrophages and two additional new subgroups that were confirmed by flow cytometry and immunohistology. Thus, our study has characterized diverse subpopulations of macrophages in resting mouse SKM.

A recent study comprehensively evaluated mouse SKM (*Wang et al., 2020*) and identified five clusters that largely overlapped with our findings. For example, the 'CD209,' 'CCR2,' and 'proliferating' clusters were very similar to our Cl0, Cl2, and Cl9, respectively (*Figure 1* and *Table 1*). Moreover, the expression of M2-like markers (e.g. *Lyve1*, *Mrc1*, *Folr2*, and *Cd163* mRNAs) suggested that the macrophages in 'unspecified cluster 0' are equivalent to our Cl1 macrophages, which also expressed many M2-like genes, although at lower levels than our Cl0. Excluding Cl0 from the comparison allowed us to identify M2-like features of the Cl1 (*Table 1*, *Supplementary file 1*). Furthermore, by analyzing both young and old SKM, we identified important new differences in macrophage clusters, including those associated with senescence and inflammation (Cl6 and Cl8, respectively). Gene expression patterns suggested that clusters 'CD209' and 'CCR2' resembled our LYVE1+/MHCII^lo and LYVE1−/MHCII^hi subgroups (*Figures 3 and 4*; *Wang et al., 2020*).

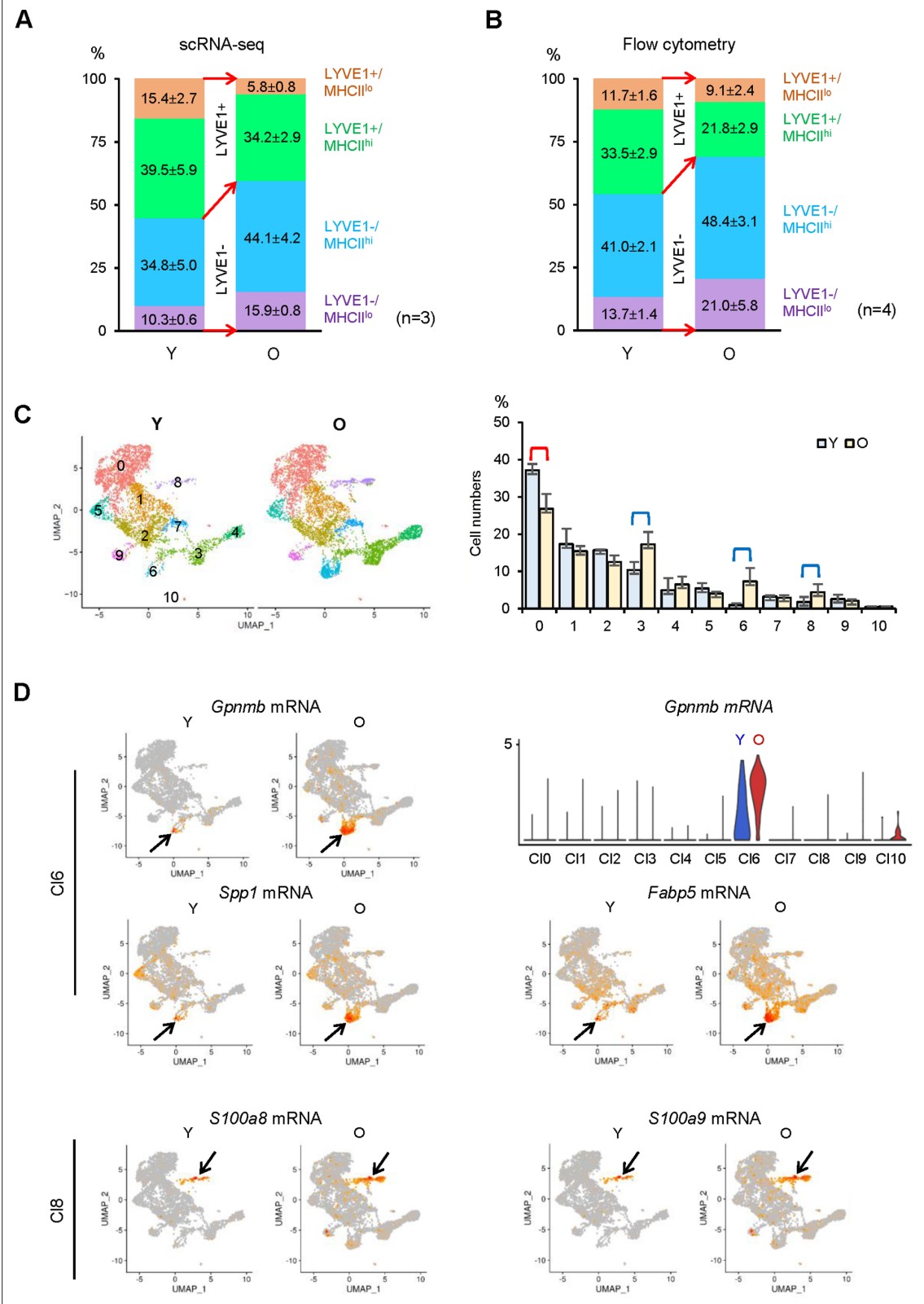

**Figure 7.** Identification of changes in macrophage subpopulations in old (**O**) relative to young (**Y**) skeletal muscle (SKM). (**A**) Single-cell RNA-sequencing (scRNA-seq) analysis showing altered numbers of LYVE1+ and LYVE1– macrophages in old SKM. (**B**) Flow cytometric analysis showing comparable changes with scRNA-seq in old SKM. (**C**) Changes in macrophage numbers in unsupervised Cl0, 3, 6, and 8. (**D**) Top, UMAP plots showing *Gpnmb*, *Spp1*

*Figure 7 continued on next page*

*Figure 7 continued*

and *Fabp5* mRNAs in old (O) and young (Y) SKM (arrow, Cl6); violin plot representing *Gpnmb* mRNA (number of macrophages and expression levels) in the different clusters. Bottom, *S100a8* and *S100a9* mRNAs in O and Y SKM (arrow, Cl8).

The online version of this article includes the following figure supplement(s) for figure 7:

**Figure supplement 1.** Changes in genes expressed by macrophages from old skeletal muscle (SKM).

**Figure supplement 2.** Biological replicates of expression patterns in young and old skeletal muscle (SKM) macrophages.

Among the supervised four subgroups, the new LYVE1+/MHCII^hi subgroup showed both M1- and M2-like gene expression patterns and functional capabilities (*Figure 3C* and *Figure 4A*). We hypothesize that this subgroup may have distinct functions or may have the potential to shift to M2-like LYVE1+/MHCII^lo or M1-like LYVE1−/MHCII^hi subgroups depending on surrounding conditions. The gene expression heat map showed that LYVE1+/MHCII^hi macrophages express features of both LYVE1+/MHCII^lo and LYVE1−/MHCII^hi, but these patterns are not prominent (*Figure 3B*). In flow cytometric analysis, LYVE1+/MHCII^hi macrophages spanned two distinct cell clusters, LYVE1+/MHCII^lo and LYVE1−/MHCII^hi (*Figure 4A*), possibly suggesting that LYVE1+/MHCII^hi macrophages represent an intermediate stage, even if they stand alone as an independent population (*Figure 4B*). The function of LYVE1+/MHCII^hi macrophages relative to LYVE1+/MHCII^lo and LYVE1^lo/MHCII^hi macrophages requires further study.

By contrast, the new LYVE1−/MHCII^lo subpopulation, which clearly separated from the other three subgroups by flow cytometric analysis (*Figure 4A*), was predicted to have a more distinct 'killing' capacity and may be directly implicated in innate immunity. In phagocytosis assays, the LYVE1−/MHCII^lo subgroup showed fewer active macrophages (*Figure 5A and B*), but those that were active had strikingly greater phagocytic capacity compared to the other three subgroups (*Figure 5D*). Unbiased further clustering suggested that this specific subgroup consists of strong (SubCl0) and weak (SubCl1-5) phagocytic subclusters (*Figure 5—figure supplement 2*), consistent with phagocytosis assays. Importantly, *Ly6c* mRNA, known to be highly expressed in circulating monocytes (*Wolf et al., 2019*), was expressed in <3% of LYVE1−/MHCII^lo and the other subgroups (not shown), while CD11c, a dendric cell (DC) marker (*Singh-Jasuja et al., 2013*), and CD49 and CD122, candidate markers for lymphoid lineage natural killer (NK) cells (*Nabekura and Lanier, 2016*), were not detected in LYVE1−/MHCII^lo or the other subgroups (*Supplementary file 3*). These data strengthen the view that LYVE1−/MHCII^lo macrophages are distinct from circulating monocytes or DC and NK cells. Additional studies are also needed to characterize the function of LYVE1−/MHCII^lo subgroup in SKM.

Our study further revealed aging-related expression changes in macrophages in SKM. Overall, LYVE1+ macrophages were less abundant, and LYVE1− macrophages were more abundant in aged SKM (*Figure 7A and B*). Consistent with these observations, *S100a8* and *S100a9* mRNAs, encoding proinflammatory biomarkers, were significantly elevated in macrophages from aged SKM. Unlike neutrophils, macrophages were reported to express S100A8 and S100A9 at low levels in the absence of stimulation (*Hessian et al., 1993*; *Wang et al., 2018*). Often forming heterodimers, S100A8 and S100A9 serve as biomarkers for the diagnosis and therapeutic responses in inflammatory diseases like inflammatory arthritis and inflammatory bowel disease, while blocking their activity resulted in reduced inflammation in mouse models (*Wang et al., 2018*). *S100a8* and *S100a9* mRNAs were in very low abundance in macrophages from young SKM but were strikingly more abundant in old SKM (*Figures 6C and 7D*). The levels of *Fabp4*, *Fabp5*, and *Il1b* mRNAs, encoding additional proinflammatory proteins, were also upregulated in macrophages from old SKM (*Figures 6C and 7D*). This finding is important because macrophage-derived FABP4 and FABP5 were shown to promote a proinflammatory state in the vasculature during atherosclerosis development (*Babaev et al., 2011*; *Furuhashi et al., 2007*; *Makowski et al., 2001*), in keeping with the proinflammatory status of old SKM. We propose that the expression levels of S100A8 and S100A9 in macrophages can be essential indicators of the inflammatory status of SKM, and possibly other tissues (*Wang et al., 2018*). Several markers of senescence and aging, including *Gpnmb* and *Spp1* mRNAs (*Pazolli et al., 2009*; *Suda et al., 2021*; *Suda et al., 2022*), were also elevated in old SKM macrophages (*Figure 6A–C*), suggesting the presence of senescent macrophages. We also found increased expression of mRNAs encoding antioxidant enzymes in old SKM macrophages, possibly reactive to elevated reactive oxygen species (ROS) in aged SKM (*Jackson and McArdle, 2011*).

By contrast, several mRNAs encoding neutrophil and monocyte/macrophage chemoattractants (*Deshmane et al., 2009*; *Girbl et al., 2018*) were expressed in lower amounts by old SKM macrophages (*Figure 6B and C*). In pathological conditions, like injury or infection, neutrophils are the earliest effector cells to infiltrate into the injury site followed by monocytes/macrophages (*Forcina et al., 2020*). At the same time, it is well known that injury repair and regeneration are slower in old SKM, perhaps due to a delay in leukocyte infiltration at early stages and to reduced CCAAT enhancer-binding protein β function toward regeneration after muscle injury (*Blackwell et al., 2015*). Thus, the reduced production of chemoattractants in macrophages may contribute to the delayed repair of older SKM.

Finally, unsupervised classification identified specific macrophage clusters significantly affected during SKM aging, particularly Cl6 and Cl8. *Gpnmb* mRNA, encoding the senescent membrane marker GPNMB (*Suda et al., 2021*; *Suda et al., 2022*), was concentrated in Cl6 and was significantly elevated in old SKM macrophages; similarly, senescence- and aging-related *Spp1* mRNA and lipid transporter *Fabp5* mRNA were highly enriched in Cl6 in old SKM macrophages (*Figures 6A, C and 7D*). On the other hand, *S100a8* and *S100a9* mRNAs were highly concentrated in Cl8 and significantly elevated in the old (*Figures 6A, C and 7D*). Thus, unsupervised clustering identified distinct subpopulations specifically altered during aging.

In closing, aging impacts all tissues and organs. Intrinsic and extrinsic factors, including DNA damage, endoplasmic reticulum stress, mitochondrial dysfunction, and a systemic inflammatory environment in aged individuals, inevitably affect the characteristics of macrophages (*van Beek et al., 2019*). A recent study suggested that macrophages from old SKM contributed to axonal degeneration and demyelination in the neuromuscular junction, and depletion of macrophages led to increased muscle endurance (*Yuan et al., 2018*). We propose that the age-associated SKM macrophage gene expression patterns identified here represent an important first step toward elucidating how macrophage subpopulations influence the pathophysiology of old SKM.

# Materials and methods

## Collection of SKMs from young and aged C57BL/6JN mice

All mouse work was done under an Animal Study Proposal (ASP #476-LGG-2023) that was reviewed and approved by the Animal Care and Use Committee of the National Institute on Aging (NIA), National Institutes of Health (NIH). Young (Y, 3 m.o.) and aged (O, 22–24 m.o.) male and female inbred C57BL/6JN mice were purchased from the NIA aged rodent colony (https://ros.nia.nih.gov/). The mice were sacrificed, and all hind limb muscles, including quadriceps, hamstring, gastrocnemius, soleus, and tibialis anterior muscles, were harvested. Collected samples were directly used for mononuclear cell isolation or frozen in isopentane chilled by liquid nitrogen and stored at –80°C for immunohistology.

## Mononuclear cell isolation from SKM

Tendons, blood vessels, and fat tissues were removed under a dissection microscope. Muscle tissues were finely chopped and minced using dissection scissors to form a slurry. For scRNA-seq analysis, we isolated mononuclear cells with Miltenyi's SKM dissociation kit (#130-098-305) with GentleMACS Octo Dissociator (#130-096-427), as described previously (*Krasniewski et al., 2022*). For further flow cytometric analysis, we also used an established method (*Liu et al., 2015*) with slight modifications. Briefly, the muscle slurry was digested with 1000 U/mL Collagenase type II (Gibco, Cat# 17101015) in 10 mL of complete Ham's F-10 medium (Lonza, Cat# BE02-014F) for 70 min with 70 rpm agitation at 37°C. Partially digested muscles were washed in complete Ham's F-10 medium and centrifuged at 400 rcf speed for 5 min, and cell pellet with 8 mL of the remaining suspension (pellet 1) was collected; 42 mL of the supernatant was collected in two tubes (21 mL each) that were filled up to 50 mL with Ham's F-10 media and centrifuged again at 500 rcf for 8 min, and the pellet (pellet 2) was collected. Pellet 1 was subjected to a second round of digestion in 1 mL of 1000 U/mL Collagenase type II and 1 mL of 11 U/mL Dispase II (Thermofisher, Cat# 17105041) along with the 8 mL of the remaining cell suspension, for 20 min with 70 rpm agitation, at 37°C. Digested tissues were aspirated and ejected slowly through 10-mL syringe with 20-gauge needle followed by washing in complete Ham's F-10 media at 400 rcf for 5 min. The supernatant was collected and centrifuged again at 500 rcf for 8 min,

and the pellet obtained (pellet 3) was pooled with the pellet 2 above. The suspension of pellets 2+3 was filtered through 40-μm cell strainer (Fisher scientific, Cat # 22363547), followed by final wash in complete Ham's F-10 medium. Cell pellets were resuspended in 1 mL complete Ham's F-10 medium. Cell counting was performed using trypan blue (Invitrogen, Cat# T10282) at a 1:1 ratio in Countess cell counting chamber slides (Invitrogen, Cat# C10228) using Countess II FL Automated Cell Counter (Invitrogen).

## Flow cytometric analysis and FACS

Flow cytometric analysis and CD11b+ cell sorting by FACS for scRNA-seq were described in detail in our previous report (*Krasniewski et al., 2022*). For further flow cytometric validation studies and RT-qPCR analysis, mononuclear cell suspensions were incubated with BD Horizon Fixable Viability Stain 780 (FVS780, BD Biosciences, Cat# 565388, dilution: 1:4000) in PBS (Ca+ and Mg+ free, Thermofisher) for 30 min at 4°C in the dark. Fc receptors were blocked using TruStain FcX (anti-mouse CD16/32) Antibody (Biolegend, Cat# 101320, Clone: 93, dilution 1:1000) for 5 min at 4°C in FACS staining buffer (1% BSA and 10 mM EDTA in Miltenyi's Auto MACS Rinsing Solution). For macrophage sorting, mononuclear cells were further stained in FACS staining buffer for 40 min at 4°C in the dark, with fluorochrome conjugated antibodies specific to mouse as indicated: BUV395 Rat anti-mouse CD45 (BD Biosciences, Cat# 564279, Clone: 30-F11, dilution: 1:100), PE anti-mouse/human CD11b Antibody (Biolegend, Cat# 101208, Clone: M1/70, dilution: 1:100), PE/Cyanine7 anti-mouse F4/80 Antibody (Biolegend, Cat# 123114, Clone: BM8, dilution: 1:40), Brilliant Violet 711 anti-mouse I-A/I-E Antibody (Biolegend, Cat# 107643, Clone: M5/114.15.2, dilution: 1:40), and APC Rat Anti-Mouse Lyve1 Antibody (Thermofisher, Cat# 50-0443-82, Clone: ALY7, dilution: 1:20; see *Supplementary file 5* for a full list of antibodies). Stained cells were fixed using BD Cytofix Fixation buffer (BD Biosciences, Cat# 554655) for 20 min on ice in the dark for analysis (but not for sorting). Compensation matrices were created using single color controls prepared using COMPtrol Kit, Goat anti-mouse Ig (H&L) coated particles, with negative and high in separate vials (Spherotech, Cat# CMIgP-30–2 K), combining one drop from each vial in equal ratio. Gating was based on FMO (fluorescence minus one) controls for each experiment. The cells were acquired on a BD FACSAria Fusion (BD Biosciences) instrument and analyzed with Flowjo software (Tree Star, Inc).

## Macrophage scRNA-seq by 10× Genomics

Macrophages isolated from three 3 m.o. and three 23 m.o. C57BL/6JN male mice (biological triplicates) were stained with CD11b antibody and isolated by FACS analysis. Given that the lengthy collection protocol made it impossible to process all the mice on the same day, we isolated cells in three consecutive weeks: from two young mice (Y1 and Y2) the first week, from two old (O1 and O2) the second week, and from one young mouse (Y3) and one old mouse (O3) the last week. Isolated SKM macrophages were immediately subjected to single-cell library construction without culture to minimize differences related to batch effects. Single-cell libraries were prepared with 10× Genomics Chromium Single Cell 3′ Reagent Kits v3 (10× Genomics Cat# PN-1000092) with Chip B (10× Genomics, Cat# PN-1000073) following the manufacturer's protocol. Briefly, 5000–10,000 single macrophages were used for GEM (Gel Bead-in-Emulsion) generation. The cDNAs were then synthesized, and their qualities were assessed on the Agilent Bioanalyzer with High-Sensitivity DNA kit (Agilent Cat# 5067–4626). cDNAs were then used for library preparation and the quality of the final libraries assessed on the Agilent Bioanalyzer with DNA 1000 kit (Agilent, Cat# 5067–1504). The libraries were sequenced with an Illumina NovaSeq 6000 sequencer with a mean depth of ~80,000 (70,876–156,962) RNA-seq reads per cell, corresponding to ~2000 (2027–2256) genes per cell. The numbers of cells from each mouse successfully sequenced and subjected to statistical analysis are as follows: Y1, 3730; Y2, 3325; Y3, 2033 and O1, 3391; O2, 5338; O3, 4097. RNA-seq data were deposited in GEO with identifier GSE195507.

## scRNA-seq data analysis

scRNA-seq samples were demultiplexed and mapped to the mm10 mouse reference genome using the Cell Ranger software version 3.0.2 (10× Genomics). Further analysis of the matrices of read counts obtained was carried out in R (version 4.1.3) with the Seurat package, version 4.1.0 (*Hao et al., 2021*), using default parameters in all functions, unless specified otherwise. To exclude empty

droplets, poor-quality cells, and potential doublets from downstream analysis, quality control filtering was applied for each sample, which removed cells containing more than 7.5% mitochondrial genes, cells expressing <300 or >7000 transcripts, and below 500 or above 60,000 counts. Genes that were detected in less than 10 cells were eliminated from the analysis. Cells expressing *Itgam* (*Cd11b*) and *Adgre1* (F4/80) mRNAs, two key macrophage markers, were subjected to further analyses.

Each sample was normalized with the LogNormalize method, and the top 2000 variable genes were selected with the FindVariableFeatures function. The SelectIntegrationFeatures function was applied to find shared variable features across the samples, and the FindIntegrationAnchors function was used to identify inter-sample anchors for integration. Then, the samples were integrated with the IntegrateData function, scaled, and subjected to principal component analysis (PCA).

For supervised cluster analysis, the macrophage dataset was divided into four cell subgroups based on the log-normalized expression values of *Lyve1 and H2-Ab1* (MHCII) mRNAs, as follows: LYVE1+/MHCII$^{lo}$ (*Lyve1* >0 and *H2-Ab1* <2), LYVE1+/MHCII$^{hi}$ (*Lyve1* >0 and *H2-Ab1* ≥2), LYVE1−/MHCII$^{hi}$ (*Lyve1* ≤0 and *H2-Ab1* ≥2), and LYVE1−/MHCII$^{lo}$ (*Lyve1* ≤0 and *H2-Ab1* <2). For unsupervised cell clustering, a shared nearest neighbor graph was generated with the FindNeighbors function (using the first 30 principal components) and clustered with Louvain algorithm in the FindClusters function with a resolution of 0.3. To visualize and explore cell clusters in a two-dimensional space, the Uniform Manifold Approximation and Projection (UMAP) analysis was performed using the first 30 principal components, as determined by the ElbowPlot method. To identify subpopulations of LYVE1−/MHCII$^{lo}$ cells, the analysis was rerun on the LYVE1−/MHCII$^{lo}$ subgroup, and clusters were visualized with resolution set to 0.3.

Differentially expressed marker genes for each cluster were identified with FindAllMarkers function, and the FindMarkers function was used to find differentially expressed genes across conditions. Those mRNAs that were expressed in at least 25% of cells per cluster were considered for differential gene expression analysis among clusters. mRNAs were defined as differentially expressed if they had an absolute fold change >1.5 and adjusted p-value<0.01. All R processing scripts are included in *Supplementary file 6*.

Functional annotation of the differentially expressed genes was performed using the web-based tool g:Profiler (*Raudvere et al., 2019*) (https://biit.cs.ut.ee/gprofiler/gost). The analysis was done with differentially expressed genes in corresponding subpopulations with 'g:SCS threshold' as a 'significance threshold' and 0.05 as the 'user threshold', and functional terms for 'GO biological process' were collected. In addition, we used 14,542 genes detected from young and old macrophages in our scRNA-seq analysis as the background gene set for GO annotation.

## RT-qPCR analysis

For RT-qPCR analysis, CD11b+/F4/80+/LYVE1+ and CD11b+/F4/80+/LYVE1− macrophages were isolated by FACS. Sorted LYVE1+ and LYVE1− macrophages were lysed with lysis buffer (RNeasy Mini Kit, Qiagen, Cat# 74104) and stored at –80°C. RNA was then isolated with a QIAcube (Qiagen) instrument following the manufacturer's protocol, using a column for RNase-Free DNase I (Qiagen, Cat# 79254) digestion. The quality of isolated RNAs was assessed on the Agilent TapeStation with RNA Screen Tape (Agilent, Cat# 5067–5576). RT was performed by synthesizing cDNAs from the LYVE1+ and LYVE1− mRNAs with the Superscript III First-Strand Synthesis System (Invitrogen, Cat# 18080051), and qPCR amplification was carried out using ready-to-use Taqman probe/primer sets (Applied Biosystems) to detect expression levels for *Lyve1* (*Mm00475056_m1*), *Folr2* (*Mm00433357_m1*), *Cd209f* (*Mm00471855_m1*), *Fcna* (*Mm00484287_m1*), *Timd4* (*Mm00724713_m1*), *Mrc1* (*Mm01329362_m1*), *Igf1* (*Mm00439560_m1*), *Ang* (*Mm01316661_m1*), *Il1b* (*Mm00434228_m1*), and *Gapdh* (*Mm99999915_g1*) mRNAs. Two biological replicates (n=2 per replicate) were used for the LYVE1+ and LYVE1− macrophages and assayed in triplicate. The relative RNA levels were calculated after normalizing to *Gapdh* mRNA using the $2^{-\Delta\Delta Ct}$ method, and the data were analyzed for significance using Student's *t*-test.

## Phagocytosis assays

Macrophages were isolated from the hind limb muscles of C57BL/6JN male mice as described above. Mononuclear cells from three animals were pooled for each set of experiments, and cells were aliquoted for necessary treatment conditions and technical replicates. Three biological replicates

(total nine mice) were analyzed. The phagocytic activity of macrophages was measured by red fluorescence from pHrodo *E. coli* bioparticles (Invitrogen, Cat# P35361). Briefly, $6\times10^6$ macrophages were resuspended in 200 µL of Ham's F-10 complete media (Lonza, 12–618 F) containing 10% horse serum (Gibco, 16050114) for each sample. Aliquots of 20 µL of pHrodo *E. coli* bioparticles, resuspended in live-cell imaging buffer (1 mg/mL, Invitrogen, Cat# A14291DJ) and sonicated for 2 min × 3, with 2 min intervals on ice between each sonication, were added to each cell tube, including appropriate FMO control tubes. Cell suspensions were gently and thoroughly mixed to ensure a homogenous distribution of the *E. coli* bioparticles. One set of samples was immediately transferred to a $CO_2$ incubator for 2 hr at 37°C, and another set (negative control) was incubated on ice for 2 hr. After incubation, cells were washed with live cell imaging solution at 400 rcf for 5 min, followed by another wash with PBS. All steps were performed in the dark.

After the phagocytosis assay, cells were stained with viability dye followed by primary antibody staining as described above. Fluorochrome-conjugated antibodies used for staining the cells are as follows: BUV395 Rat anti-mouse CD45, PE-Cyanine7 anti-mouse/human CD11b Antibody, BUV737 Rat anti-mouse F4/80, Brilliant Violet 711 anti-mouse I-A/I-E Antibody, APC Rat anti-mouse LYVE1 Antibody. The cells were acquired on a BD FACSAria Fusion instrument on the same day and analyzed with Flowjo. For all the samples, including controls, CD11b+/F4/80+macrophages were further categorized as high (Hi, >$10^5$), medium (Med, $10^4$–$10^5$), low (Lo, $10^3$–$10^4$), and negative (Neg, <$10^3$) intensity groups based on their ability to engulf labeled bacteria. The relative phagocytosis levels for each group were calculated using gMFI. For statistical analysis, we performed a Shapiro-Wilk test (*Mishra et al., 2019*) first to assess if our data were normally distributed (GraphPad Prism 8). We found that all data shown in *Figure 5B and D*, and *Figure 4—figure supplement 1B*, and *Figure 5—figure supplement 1B* were normally distributed (not shown). Therefore, we performed parametric tests, one-way ANOVA (Dunnett's multiple comparisons test) for *Figure 5D*, and two-way ANOVA (Sidak's multiple comparisons test) for *Figure 5B* and *Figure 4—figure supplement 1B*, and *Figure 5—figure supplement 1B* using GraphPad Prism 8.

## Efferocytosis assay

To study the engulfment of apoptotic cells by SKM macrophages, hind limb muscles from three male mice were combined as one biological replicate, followed by digestion to generate a mononuclear cell suspension as mentioned above, and pooled for study of phagocytic cells. To generate apoptotic cells, Jurkat T cells, cultured in RPMI 1640 (Thermo Fisher, Cat# 11875–093) with 10% heat-inactivated FBS (Thermo Fisher, Cat# 10438025) at 37°C, 5% $CO_2$ 10 mL were collected from a cell culture flask, washed with PBS, pelleted gently, and resuspended in 1 mL PBS. For labeling Jurkat cells, CFSE (CellTrace CFSE Cell Proliferation Kit, Thermo Fisher, Cat# C34554) was added to cells at final 5 µM concentration and incubated at 37°C for 20 min; 10 mL of RPMI 1640 (10% heat-inactivated FBS) was then added, mixed by vortexing, and further incubated at 37°C for 5 min. After washing, cells were resuspended in 1 mL serum-free RPMI 1640 medium. Apoptosis was induced by treatment with 1 µM staurosporine (Millipore Sigma, Cat# 19–123) for 5 hr at 37°C, 5% $CO_2$, followed by washes in RPMI 1640 (10% heat-inactivated FBS) and resuspension in 1 mL RPMI 1640 (10% heat-inactivated FBS) for use in efferocytosis assays. Mononuclear cells from SKM and apoptotic Jurkat T cells were counted and combined at a 1:1 ratio in 2 mL RPMI 1640 (10% heat-inactivated FBS) and incubated for 18 hr at 37°C, 5% $CO_2$. Mononuclear cells from SKM without Jurkat cells were used as controls. After incubation for 18 hr, cells were assayed by flow cytometry, as explained above.

## Immunofluorescent staining of macrophages in mouse SKM

Frozen sections from rectus femoris muscle from 3 m.o. C57BL/6 J mice were cut, fixed in cold acetone, and subjected to regular double immunofluorescent staining or double TSA staining (Tyramide SuperBoost kit, Thermo Fisher, Cat# B40932) as performed previously (*Cui et al., 2019*). Primary antibodies recognizing LYVE1 (Abcam, Cat# ab14917, 1:200 dilution), MHCII (Invitrogen, Cat# 14-5321-82, 1:100), CD31 (Millipore, Cat# MAB1398Z, 1:100), and TUBB3 (Biolegend, Cat# 801201, 1:200) worked well for regular immunostaining. Secondary antibodies were used for LYVE1 (Invitrogen, Cat# A-11012, 1:1000 dilution) and MHCII staining (Invitrogen, Cat# A-11006, 1:1000 dilution). Detection of CD11b (Santa Cruz, Cat# sc-1186, 1:50 dilution) required TSA staining. To identify LYVE1+/MHCII$^{hi}$ macrophages in SKM, we carried out three sets of double staining: LYVE1 with MHCII

by regular immunofluorescence staining, CD11b with LYVE1, and CD11b with MHCII by TSA staining. Micrographs were taken on a DeltaVision microscope using a 20× lens.

## Acknowledgements

This work was supported by the Intramural Research Program of the National Institute on Aging. The authors thank Marc Michel for technical assistance.

## Additional information

### Funding

| Funder | Grant reference number | Author |
|---|---|---|
| National Institutes of Health | Z01-AG000511 | Linda K Krasniewski |

The funders had no role in study design, data collection and interpretation, or the decision to submit the work for publication.

### Author contributions

Linda K Krasniewski, Papiya Chakraborty, Conceptualization, Investigation, Methodology, Writing – original draft; Chang-Yi Cui, Conceptualization, Data curation, Formal analysis, Investigation, Writing – original draft, Project administration, Writing – review and editing; Krystyna Mazan-Mamczarz, Data curation, Formal analysis, Investigation, Methodology; Christopher Dunn, Formal analysis, Investigation, Visualization; Yulan Piao, Jinshui Fan, Isabelle A Rathbun, Formal analysis, Investigation; Changyou Shi, Supervision, Investigation, Methodology; Tonya Wallace, Cuong Nguyen, Resources, Investigation; Rachel Munk, Resources, Investigation, Methodology; Dimitrios Tsitsipatis, Formal analysis, Validation, Investigation, Methodology; Supriyo De, Resources, Software, Supervision; Payel Sen, Supervision, Methodology; Luigi Ferrucci, Resources, Supervision; Myriam Gorospe, Conceptualization, Investigation, Writing – original draft, Project administration, Writing – review and editing

### Author ORCIDs

Chang-Yi Cui http://orcid.org/0000-0002-9856-9427
Christopher Dunn http://orcid.org/0000-0001-7899-0110
Payel Sen http://orcid.org/0000-0003-2809-0901
Luigi Ferrucci http://orcid.org/0000-0002-6273-1613
Myriam Gorospe http://orcid.org/0000-0001-5439-3434

### Ethics

All animal study protocols were approved by the NIA Institutional Review Board (Animal Care and Use Committee). (ASP #476-LGG-2024).

### Decision letter and Author response

Decision letter https://doi.org/10.7554/eLife.77974.sa1
Author response https://doi.org/10.7554/eLife.77974.sa2

## Additional files

### Supplementary files

• Supplementary file 1. Top mRNAs differentially expressed in each of 11 unsupervised clusters (Cl0-Cl10), identified after single-cell analysis using 10× Genomics (Materials and methods).

• Supplementary file 2. mRNAs highly expressed in LYVE1+ or LYVE1− macrophages.

• Supplementary file 3. Top mRNAs in each of the four supervised macrophage subgroups, LYVE1+/MHCII$^{lo}$, LYVE1+/MHCII$^{hi}$, LYVE1−/MHCII$^{hi}$, and LYVE1−/MHCII$^{lo}$.

• Supplementary file 4. Differentially expressed mRNAs in skeletal muscle (SKM) macrophages from young and old mice.

• Supplementary file 5. List of antibodies used in this study, including catalog number, company, and dilution used.

• Supplementary file 6. R processing scripts used for data analysis.

• Transparent reporting form

### Data availability

The single-cell RNA-seq analysis was uploaded to GEO with identifier GSE195507.

The following dataset was generated:

| Author(s) | Year | Dataset title | Dataset URL | Database and Identifier |
|---|---|---|---|---|
| Krasniewski LK, Chakraborty P, Cui CY, Mazan-Mamczarz K, Dunn C, Piao Y, Fan J, Shi C, Wallace T, Nguyen C, Rathbun IA, Munk R, Tsitsipatis D, De S, Sen P, Ferrucci L, Gorospe M | 2022 | Single-cell analysis of skeletal muscle macrophages reveals age-associated functional subpopulations | https://www.ncbi.nlm.nih.gov/geo/query/acc.cgi?acc=GSE195507 | NCBI Gene Expression Omnibus, GSE195507 |

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
