## [Editor Report]

In this study, Krasniewski and colleagues describe important findings leveraging single-cell transcriptomics to identify subpopulations of macrophages in the skeletal muscle of aging mice. They present solid evidence for the existence of several new resident subpopulations of skeletal muscle macrophages, spanning a range of polarization states using novel markers. Additionally, they identify a shift in relative abundances of these subpopulations with age, leading to a functional shift in inflammatory marker expression and phagocytic capacity. This work will be useful to researchers in the field of immune aging as a resource.

---

## [Decision Letter]

**Decision letter after peer review:**

Thank you for submitting your article "Single-cell analysis of skeletal muscle macrophages reveals age- associated functional subpopulations" for consideration by *eLife*. Your article has been reviewed by 3 peer reviewers, one of whom is a member of our Board of Reviewing Editors, and the evaluation has been overseen by Carlos Isales as the Senior Editor. The reviewers have opted to remain anonymous.

Essential revisions:

The reviewers and I discussed the manuscript, and we believe that while some major revisions are needed, this manuscript could be appropriate for *eLife* after major revisions. The most salient are summarized below:

1. The choice of a supervised clustering approach for macrophage heterogeneity analysis was puzzling to reviewers, especially with the choice of using noncanonical markers – the rationale for not using an unsupervised approach was not well explained. In addition, the chosen non-standard nomenclature is confusing, and it is recommended to avoid shorthand to improve the reading experience. In a revised manuscript, the authors should start with an unsupervised approach, as is the gold standard of the field, which can then reveal specific markers that may be used to segregate functional subpopulations.

2. The manuscript needs to provide additional information about both wet and dry methods used to generate results for improved reproducibility: (i) better clarification of potential batch processing, (ii) more information on the functional enrichment analysis, (iii) support for using parametric tests (i.e. tests for normality), (iv) extensive information on package and software versions used for analysis, (v) catalog details for flow antibodies and information on the sorting schemes, (vI) deposition of all R processing scripts, etc.

3. Information about the provenance of the macrophages in the tissue (i.e. associated to blood vessels, inside the parenchyma, etc.) in relation to the diverse phenotypes identified using scRNAseq.

4. Many conclusions need to be toned down as most of the functional information is derived from genomic annotation and not functional assays (i.e. the "wound healing" discussion in the absence of healing assays or of efferocytosis assays) to reflect the degree of evidentiary support. It is also noted that lower numbers of DE genes in scRNAseq datasets is a common caveat of the method that should not be overinterpreted.

5. The reviewers were concerned about the lack of consideration of sex as a biological variable. It was discussed that, in the absence of scRNAseq data in female animals, either some functional experiments should be performed in females to confirm the broader applicability of results, or the male-specific nature of the paper should be explicitly discussed.

In general, all discussed data should be shown and conclusions should not reach beyond that which is directly supported by data unless explicitly stated to be speculation.

*Reviewer #1 (Recommendations for the authors):*

1. Aging and immunity are both very sex dimorphic. Since the authors profiled exclusively male animals, it will be important to explicitly discuss how these results may differ in female animals in the Discussion section.

2. When describing results from the flow cytometry-based phagocytosis assay (Figure 4), the authors find that LnHl macrophages are phagocytic to a lower proportion than other described subgroups (~49% compared to >85%), although the phagocytic macrophages show strikingly higher levels of phagocytosed cargo by MFI analysis. Although we agree that the significance of this is unclear with the current evidence, this suggests further heterogeneity in the LnHl group. It would thus be important to try to use unbiased SNN clustering of LnHl macrophages (not just of all macrophages as in Figure 6) and identify potential subpopulations that may explain this functional heterogeneity using the generated scRNAseq data (for instance as relating to phagocytosis-gene related mRNAs).

3. Based on experimental flow/description, it sounds like young and old samples may not have been processed in parallel, which may be problematic due to the known impact of batch effects in genomics. Can the authors clarify and explicitly discuss whether batching may be a problem?

4. There needs to be additional provided information about some of the bioinformatic tools and/or analyses.

a. Although the authors generally provide information about software/package versions or dates of access, some are missing (e.g. R, g:profiler). This needs to be updated for reproducibility.

b. Although a GO analysis with g:profiler is described in the text and figures, the method is not described in the method section. Since the nature/use of background lists in functional enrichment analysis is crucial, the authors should clarify the list of genes used as background for enrichment analysis (ideally all detected/expressed genes), as well as the FDR threshold for considering a term significantly enriched. A supplementary table with all enriched terms would also be invaluable.

c. For long-term reproducibility, it would be important to either deposit all R scripts to a public repository such as github or provide it as a supplemental archive to accompany the manuscript.

d. In the methods, the authors mention using Student's t-tests, but not tests to verify that data was normally distributed before use of the t-test. Please include the reference to any normality test used, or, if normality of data cannot be verified, please update to use non-parametric tests.

5. Please provide catalog numbers for the antibodies used in the flow phagocytosis assay, as is needed for reproducibility (methods, page 20).

*Reviewer #2 (Recommendations for the authors):*

1) Pg5 the authors rely on a limited number of markers to determine the macrophage polarization status (M1 vs M2) of the different subsets of macrophages characterized in the study. Can the authors investigate more exclusive markers (PMID: 26699615) to determine if MHCII and Lyve macrophage subsets are indeed skewed to one state or the other?

2) Lp+ macrophages were described in the study to express a transcriptomic program characterized by M2-like gene program involved in wound repair and healing. Thus, beyond investigating the phagocytic capacity of the macrophage using labeled *E. coli*, can the authors test the uptake of apoptotic cells using an efferocytosis assay, which seems more relevant for wound repair.

3) Pg7 "We found that MHCII mRNAs (encoding H2-Ab1, H2-Eb1) divided SKM macrophages into two groups, MHCII-high (Hh) and MHCII-low (Hl) in single-cell profiling analysis." Please provide a figure reference.

4) Pg 8, in the immunostaining are different subsets of macrophages localized to different sections of muscle tissues? i.e. more associated with endothelial cells, muscle cells or innervated areas?

5) Can the authors provide more insight into both old and young skeletal muscle to determine if an absolute number of macrophages change during aging (for example via flow cytometry) and also to determine using transcriptomics if the subpopulations of macrophages in the study are resident vs non-resident macrophages.

6) The study lacks analysis of skeletal muscle macrophages in female young and old mice, and would be more informative if they were included in the study to determine any sex differences. Perhaps qPCR or flow cytometry analysis of the four major subsets can be investigated in female mice, since performing single cell would be cost prohibitive.

7) Pg 11. "Cdk1 and Top2a mRNAs were expressed in an even lower number of macrophages (data not shown)." I am confused by data not shown since the data appears to be in table. If not please show.

*Reviewer #3 (Recommendations for the authors):*

1) Manuscript provides no functional insights of how changes in macrophages affect muscle-physiology/pathology in young and aged mice. Conclusion about functions are based on associations.

2) Please clarify what the percentage of macrophages in muscle in comparison to other CD45+ cells is.

3) Please clarify wheter the analyzed myeloid cells are present in vessels or in muscle parenchyma. How that affects muscle function is also unclear.

4) While the scRNAseq of skeletal muscle macrophages reveals interesting findings about their diversity and how aging affects their transcriptomic profile, the authors chose an unusual approach to analyze the data by defining subsets based on extracellular marker expression (Lyve1 and MHCII) and not transcriptomic profile. By starting with a supervised approach, the authors have missed an important part of their data regarding the high diversity of skeletal muscle macrophages, that cannot only be described through MHC-II and Lyve1 expression. The expression of extracellular markers does not necessarily allow to define functionally distinct subsets. A deeper and more detailed unsupervised analysis is required for scRNA seq data. Moreover, as a starting point, the clusters should be defined based on gene expression and then propose candidate markers to characterize the subpopulations and their functional properties

5) Several conclusions are not supported by the data – e.g.

a) Naming Lyve1+ and Lyve1- macrophages "healing" and pro-inflammatory" is premature as this has not been functionally tested.

b) Unless the authors have data showing the general health of the animals used for the scRNAseq, the low number of differentially expressed genes between young and old shouldn't be interpreted as a result of a "healthier" status of the aging cohort. It is rather a common caveat of scRNAseq that lacks sequencing depth. A bulk RNAseq of specific sorted population that changes in aging may be important.

6) Characterization of macrophages with polarization markers is inadequate and based on old literature. Hence, not very informative.

7) The unsupervised analysis of the scRNAseq lacks depth. Each cluster should be described independently of whether they belong to LpHl, LpHh, LnHh and LnHl. This analysis should come as the first figure. For example, do not dismiss the possibility of non-macrophage clusters as cells were only sorted on CD11b marker. Cluster 6 could be a granulocyte cluster.

8) Overall, the logic behind the direction taken for the analysis of the data is unclear:

– It is not clear why the authors chose to perform a scRNAseq. If the goals of the authors were to study macrophages subsets based on extracellular markers, sorting these subsets, and performing a bulk RNAseq would have been more adequate and would have provided and better sequencing depth.

– The choice of MHC-II and Lyve1 markers to divide macrophage subset seems arbitrary. It is not clear why the authors chose to use markers described for lung macrophages specifically.

– The data showing phagocytic capacities are interesting. Though, it is not clear why the authors chose to investigate this function. Is phagocytosis relevant to the physiological function of skeletal muscle tissues? Are there other functions highlighted in the GO annotations that can be tested? (endocytosis, inflammatory response, antigen presentation)

9) It is difficult to evaluate the solidity of the data. For quantification of populations by flow cytometry analysis and using the scRNAseq data, we suggest adding histograms representing n numbers and statistical significance when comparing the abundance of populations (Figure 3D, S3C, 5A, 6A). Moreover, instead of using tables to depict differentially expressed genes, heatmaps or volcanoplots are recommended (Figure 5C, S3C).

Figure 1C and Figure 2: Ln cells appear to be multiple clusters – with same features? How is that possible?

Figure 1D: The bioinformatic analyses used to characterize Lp/Ln cells lacks statistical validity.

Figure 3D: biological purpose to reveal new insights is lacking.

Figure 3E; Fails to provide spatial clarity of localization in muscle.

Figure 4: phagocytic analyses are superficial and fail to provide new insights.

Figure 6: is the most important figure. However, it is poorly put together in terms of data analyses and presentation. Identity of cluster 5 and 6 which change with age is the main finding – but their relevance is unknown

[Editors’ note: further revisions were suggested prior to acceptance, as described below.]

Thank you for resubmitting your work entitled "Single-cell analysis of skeletal muscle macrophages reveals age-associated functional subpopulations" for further consideration by *eLife*. Your revised article has been evaluated by Carlos Isales (Senior Editor) and a Reviewing Editor.

The manuscript has been improved but there are some remaining issues that need to be addressed, as outlined below:

Notably, after a new round of review/discussion, the reviewers felt like the issue of batching in the data needed further information/analyses to make sure that the reported results are not the result of batch and biology being confounded in 2 out of the 3 replicates for each age.

Upon discussion, the reviewers thought that this could be addressed with:

1. Doing an analysis paralleling the main manuscript but using only the 2 samples that were processed "unbatched" (i.e. the samples where 1 young and 1 old mouse were processed in parallel), and only these. If the main results of the study are conserved in this unbatched subset of the data, this would strengthen the likelihood that the batching did not grossly impact the conclusions. We would then recommend including this analysis as a supplement.

2. As highlighted by reviewer #3, it is crucial that the batching/experimental collection scheme be discussed explicitly in the manuscript.

3. Finally, please address Reviewer #1's remaining concern on the use of a background gene list for the g:profiler analysis.

Thank you!

*Reviewer #1 (Recommendations for the authors):*

Although the authors have addressed most of my concerns, some large concerns remain at this point.

1. A very large concern was revealed by their answer to one of my questions about the batchiness of the data. Indeed, the author's response revealed 3 batched: (i) only young samples, (ii) only old samples and (ii) one old and one young. Unfortunately, since batch and biological groups are confounded for groups i and ii, that data is meaningless (i.e. batch cannot be properly accounted for when it is confounded with biology). Since I understand that the authors may not be able to redo the entire experiment the way it should have been done, I believe it is imperative that all analyses also be done exclusively on batch 3 (the one where both groups were represented), to show that all results would hold in the absence of batch. The results should then be included and compared/discussed in the context of the paper as this is a big problem.

2. The authors still did not address the background list used for GO enrichment in g:profiler. This leads me to believe they used the default (all genes instead of detected genes in the dataset), which is incorrect and would lead to spurious enrichments. These analyses should be rerun with the correct background list.

*Reviewer #2 (Recommendations for the authors):*

We believe the authors have responded to the concerns of the reviewers sufficiently and the paper is significantly improved. Thus, in our opinion, the paper is suitable for publication.

*Reviewer #3 (Recommendations for the authors):*

The authors have addressed most of my prior concerns. Some issues remain, but in general given the importance of the topic, the manuscript is ready to forward in the process.

An important issue that remains unaddressed is that the scRNA analyses and cell sorting for young/old groups were done on different days. The authors responded to this issue and acknowledged this caveat, but do not describe the consequence of this on the data generation and conclusions.

Authors need to provide this information. This reviewer could not find it in the beginning of the Results section where the description of data generation is provided.

---

## [Author Response]

Essential revisions:The reviewers and I discussed the manuscript, and we believe that while some major revisions are needed, this manuscript could be appropriate for eLife after major revisions. The most salient are summarized below:1. The choice of a supervised clustering approach for macrophage heterogeneity analysis was puzzling to reviewers, especially with the choice of using noncanonical markers – the rationale for not using an unsupervised approach was not well explained. In addition, the chosen non-standard nomenclature is confusing, and it is recommended to avoid shorthand to improve the reading experience. In a revised manuscript, the authors should start with an unsupervised approach, as is the gold standard of the field, which can then reveal specific markers that may be used to segregate functional subpopulations.

We sincerely appreciate your advice after consulting the reviewers. Following your recommendation, we have reorganized our revised manuscript to start with unsupervised clustering skeletal muscle (SKM) macrophages by single-cell RNA-sequencing analysis, followed by supervised classification using membrane markers LYVE1/MHCII, and ending with the functional analysis. We agree that this approach provides a better flow and better appreciation of the complementary between the two methods for analysis of SKM macrophages.

We agree that we had not explained well why our original manuscript began with the classification of macrophages using noncanonical membrane markers. Briefly, while the traditional classification of M1 and M2 macrophages was largely based on membrane markers CD206, CD86, and CD80, LYVE1 had recently emerged as a promising membrane marker for M2 macrophages, and MHCII proteins for M1 macrophages, and therefore we examined them here. As indicated in our manuscript, for functionally subgrouping skeletal muscle macrophages, the LYVE1/MHCII combination appeared to be more informative than the traditional CD206/CD86 or CD206/CD80 combination.

Following your advice, we also changed our designation of the macrophage subgroups to follow more conventional nomenclature across the text and figures: ‘Lp’ is now LYVE1+, ‘Ln’ is LYVE1-, ‘LpHl’ is LYVE1+/MHCII^lo^, ‘LpHh’ is LYVE1+/MHCII^hi^; ‘LnHp’ is LYVE1-/MHCII^hi^; and ‘LnHl’ is LYVE1-/MHCII^lo^.

2. The manuscript needs to provide additional information about both wet and dry methods used to generate results for improved reproducibility: (i) better clarification of potential batch processing, (ii) more information on the functional enrichment analysis, (iii) support for using parametric tests (i.e. tests for normality), (iv) extensive information on package and software versions used for analysis, (v) catalog details for flow antibodies and information on the sorting schemes, (vI) deposition of all R processing scripts, etc.

We fully agree with these requests and have implemented them as follows:

(i) Better clarification of potential batch processing:

To avoid batch effects, we had adopted the following measures. For wet-lab methods, we isolated cells in 3 consecutive weeks, from 2 young mice one week, from 2 old mice on the second week, and from 1 young and 1 old mice on the third week. Single-cell libraries were prepared on the same day of each isolation using the 10x Genomics reagents and methods. We have included this information in the ‘Materials and methods’ section.

For dry-lab methods, we did not expect substantial batch effects in the data, as the sequencing of all the samples was performed at the same time, and the absence of batch effects was confirmed by careful visual inspection of cell distributions in each sample. Additionally, batch-effect corrections are part of the integration steps in the Seurat package we used in our analysis, which eliminates potential technical differences across samples (https://satijalab.org/seurat/articles/integration_introduction.html). In the revised version, we described these steps more thoroughly in the Material and Methods section and present cell alignments across all samples (Figure 1—figure supplement 1B, Figure 3—figure supplement 1B, and Figure 7—figure supplement 2).

(ii) More information on functional enrichment analysis:

The functional enrichment analysis was carried out using g:Profiler. This information is included in the revised Materials and methods section.

(iii) Support for using parametric tests (i.e. tests for normality):

In the revised Materials and methods section, we have updated the tests used for normality. We performed a Shapiro-Wilk test first to confirm the normal distribution of our data. We then performed parametric tests (one-way or two-way ANOVA). We have described these details in the revised Materials and methods section and have included the results of the Shapiro-Wilk test in our response to Reviewer 1 (below).

(iv) Extensive information on package and software versions:

In the revised Materials and methods section, we have updated the information on tests, packages, and software versions used.

(v) Catalog details for flow antibodies and information on the sorting schemes:

We have added a supplementary Table S6 in which we include catalog numbers for the antibodies used in the flow cytometry and immunofluorescence analyses. For information on the sorting schemes, we have reported details in our protocol paper, which is in press in Bio-Protocols now (and we have appended as a supplemental item in this revision). In the revised manuscript, we have stated this information more clearly, and have made clearer the sorting procedure, under ‘Flow cytometric analysis and FACS’ section.

(vi) Deposition of all R processing scripts, etc.

Following the advice of Reviewer 1, we have added a supplementary archive (Table S7) that contains all the scripts used.

3. Information about the provenance of the macrophages in the tissue (i.e. associated to blood vessels, inside the parenchyma, etc.) in relation to the diverse phenotypes identified using scRNAseq.

We appreciate this important request. To address it, we have carried out additional immunostaining to identify cells from other tissues associated with the muscle. These findings have been added to the new Figures 4B and 4C. Briefly, all macrophages were localized in the perimysium or endomysium areas (i.e., between muscle fibers, but not inside muscle fibers) as shown in Figure 4B. We further found that LYVE1+ macrophages (LYVE1+/MHCII^lo^ and LYVE1+/MHCII^hi^), but not LYVE1-/MHCII^hi^ macrophages, were closely localized with blood vessels (Figure 4C, top), similar to previous findings in lung tissue, reported by Chakarov et al., 2019. However, all macrophage subtypes were located around nerve fibers. We have revised the text to highlight these results when we describe Figure 4C.

4. Many conclusions need to be toned down as most of the functional information is derived from genomic annotation and not functional assays (i.e. the "wound healing" discussion in the absence of healing assays or of efferocytosis assays) to reflect the degree of evidentiary support. It is also noted that lower numbers of DE genes in scRNAseq datasets is a common caveat of the method that should not be overinterpreted.

We fully agree with these recommendations. We have toned down the interpretations of the data throughout the text, particularly when the relevant functional assays were not available. Regarding the lower number of differentially expressed genes in scRNAseq datasets, we appreciate the Reviewer’s excellent advice and have made explicit note of this limitation in the revised Results.

5. The reviewers were concerned about the lack of consideration of sex as a biological variable. It was discussed that, in the absence of scRNAseq data in female animals, either some functional experiments should be performed in females to confirm the broader applicability of results, or the male-specific nature of the paper should be explicitly discussed.

We agree with and appreciate these recommendations. To address them, we now clearly state that the work was carried out using male mice in the Abstract, Results, Discussion, and Materials and methods sections. In addition, to understand possible male-vs-female influences on macrophage polarization in SKM, we carried out flow cytometric analysis using macrophages from female mice. We found statistically decreases in the LYVE1+/MHCII^hi^ subgroup and increases in LYVE1-/MHCII^lo^ macrophages in females compared to males. We included these data in the supplement (Figure 4—figure supplement 1A,B). These results have encouraged us to initiate more detailed studies to understand the biological significance of the differences between male and female SKM macrophages, and we hope to report them in the near future as a separate investigation.

In general, all discussed data should be shown and conclusions should not reach beyond that which is directly supported by data unless explicitly stated to be speculation.

We appreciate and agree with this advice. We have toned down our conclusions across the text.

Reviewer #1 (Recommendations for the authors):1. Aging and immunity are both very sex dimorphic. Since the authors profiled exclusively male animals, it will be important to explicitly discuss how these results may differ in female animals in the Discussion section.

We appreciate this important reminder. We fully agree with the Reviewer and have revised the text in two ways. First, we state in the revised text (Abstract, Results, Discussion, and Materials and methods sections) that the study was done using male mice. Second, we isolated macrophages from female SKM, as suggested by Reviewer 2, and studied them by flow cytometry. Although female SKM macrophages distributed into the same 4 subgroups and these were comparable in size to those of males, the LYVE1+/MHCII^hi^ subgroup was relatively smaller and the LYVE1-/MHCII^lo^ subgroup larger than those of males. We included these data in the supplement (Figure 4—figure supplement 1A,B) and briefly described them in the Results section. In light these unexpected findings, we have begun dedicated studies to understand the biological significance of the differences between male and female macrophages in SKM, and we hope to report them in the near future as a separate investigation.

2. When describing results from the flow cytometry-based phagocytosis assay (Figure 4), the authors find that LnHl macrophages are phagocytic to a lower proportion than other described subgroups (~49% compared to >85%), although the phagocytic macrophages show strikingly higher levels of phagocytosed cargo by MFI analysis. Although we agree that the significance of this is unclear with the current evidence, this suggests further heterogeneity in the LnHl group. It would thus be important to try to use unbiased SNN clustering of LnHl macrophages (not just of all macrophages as in Figure 6) and identify potential subpopulations that may explain this functional heterogeneity using the generated scRNAseq data (for instance as relating to phagocytosis-gene related mRNAs).

We appreciate the Reviewer’s helpful observations. We performed unsupervised clustering specifically for the LnHl subgroup (now renamed LYVE1-/MHCII^lo^) and found 6 subclusters within this subgroup of macrophages. Among them, subclusters 0, 2 and 3 showed strong correlation with phagocytosis/endocytosis. Subcluster 1 showed only one endocytosis term with low negative p-value, and subclusters 4 and 5 did not show any terms related to phagocytosis or endocytosis. We believe that within the LYVE1-/MHCII^lo^ subgroup, subclusters 0, 2 and 3 represent strong phagocytotic macrophages, and subclusters 1, 4 and 5 represent weakly phagocytotic macrophages. We included the unsupervised clustering and GO annotation in Figure 5—figure supplement 2A,B, listed the genes related to phagocytosis/endocytosis in the corresponding subclusters in Extended File 4, and described these data in the Results section.

3. Based on experimental flow/description, it sounds like young and old samples may not have been processed in parallel, which may be problematic due to the known impact of batch effects in genomics. Can the authors clarify and explicitly discuss whether batching may be a problem?

We apologize for the poor description of our experiment in the original submission. We isolated macrophages from two young mice in the first week, from two old mice in the second week, and from one young mouse and one old mouse in the third week. Single-cell libraries were prepared on the same day that we performed each isolation. We have added text describing these details in the Results and the Materials and methods sections. In the revised version, we also provide the biological replicates in supplement (Figure 1—figure supplement 1, Figure 3—figure supplement 1, Figure 7—figure supplement 2) and have modified the text descriptions.

4. There needs to be additional provided information about some of the bioinformatic tools and/or analyses.

We appreciate these specific requests. We have addressed them as explained below.

a. Although the authors generally provide information about software/package versions or dates of access, some are missing (e.g. R, g:profiler). This needs to be updated for reproducibility.

Regarding the use of g:Profiler, we added more details to the information in Materials and methods section.

b. Although a GO analysis with g:profiler is described in the text and figures, the method is not described in the method section. Since the nature/use of background lists in functional enrichment analysis is crucial, the authors should clarify the list of genes used as background for enrichment analysis (ideally all detected/expressed genes), as well as the FDR threshold for considering a term significantly enriched. A supplementary table with all enriched terms would also be invaluable.

We appreciate the Reviewer’s request and fully agree that more information and clarification are needed. First, we have included a description of the GO analysis with g:profiler, including significance thresholds and user threshold, as mentioned above, in response to section (a) of this query. Second, we have included all those genes that are significantly differentially expressed in each cluster/subgroup used for GO annotation in Supplementary Files 1 through 6. Third, the GO annotation usually provides more than a hundred functional terms, many of which are general functions of macrophages shared by most macrophages, as mentioned in the revised Results section, and many represent similar functional terms. We tried to select significant but non-redundant functional terms from them and displayed them in the figures and text. In addition, as the program is frequently updated, we felt that providing the list of featured genes (rather than the very long list of enriched terms), in the Supplementary Files 1 through 6 would be more informative. Interested readers could then easily access g:profiler (freely available) and can reanalyze GO annotation with upgraded g:profiler versions in the future. If the Reviewer prefers that we proceed differently, we ask for his/her kind advice.

c. For long-term reproducibility, it would be important to either deposit all R scripts to a public repository such as github or provide it as a supplemental archive to accompany the manuscript.

We appreciate this suggestion and have provided the R scripts as a supplemental archive, uploaded with this submission (Supplementary File 7).

d. In the methods, the authors mention using Student's t-tests, but not tests to verify that data was normally distributed before use of the t-test. Please include the reference to any normality test used, or, if normality of data cannot be verified, please update to use non-parametric tests.

We appreciate these valuable comments. In the revised manuscript, we have used the ‘Shapiro-Wilk test’ program in GraphPad Prism 8 to test for normal distribution. We observed that all of our data/comparisons are normally distributed. Although we mention these results as ‘data not shown’, we have displayed the results for the benefit of the Reviewer (Author response image 1). Based on the normality test, we proceeded to perform parametric tests and used one-way ANOVA (Dunnett's multiple comparisons test) for Figure 5D, and two-way ANOVA (Šídák's multiple comparisons test) for Figures 5B, Figure 5—figure supplements 1B and 2B.

**Author response image 1. sa2fig1:** 

5. Please provide catalog numbers for the antibodies used in the flow phagocytosis assay, as is needed for reproducibility (methods, page 20).

We appreciate this request. In the new Supplementary File 6, we list the antibodies (and catalog numbers) used for flow cytometric analysis and tissue immunofluorescence staining.

Reviewer #2 (Recommendations for the authors):1) Pg5 the authors rely on a limited number of markers to determine the macrophage polarization status (M1 vs M2) of the different subsets of macrophages characterized in the study. Can the authors investigate more exclusive markers (PMID: 26699615) to determine if MHCII and Lyve macrophage subsets are indeed skewed to one state or the other?

We thank Reviewer 2 for bringing to our attention the study by Jablonski et al. (PLoS ONE, 2015). We compared our data with the data in this paper while preparing our initial manuscript. However, there were major differences in macrophage processing, as macrophages were cultured and treated in the PLoS ONE study but were not cultured or treated in our study, and also major differences in detection of the expressed transcriptome, as microarrays were used in the PLoS ONE study, RNA-seq analysis in ours. Consequently, the gene expression patterns in SKM macrophages in our analysis were quite different from those published in PLoS ONE, except for very typical M2 markers like Mrc1 mRNA and M1 markers like Il1b mRNA. We picked top M1 markers, such as Cd38, Fpr2, Gpr18, Hp, Cfb, Cxcl10, Ccr7, and Il6 mRNAs, and top M2 markers like Egr2, Myc, Cd83, Ptgs1, Flrt2, Mmp12, Fn1, Arg1, Chi3l3, and Renla mRNAs from the PLoS ONE paper, and compared them with our data. The above genes were expressed at very low levels in general in our data, and showed no clear skewed expression pattern in either supervised or unsupervised clustering. Thus, we felt that discussing this paper would not have been very informative. However, another paper that characterized SKM macrophage origin and subgroups (Wang et al., Proc. Natl. Acad. Sci. USA, 2020) with an experimental setup more similar to ours showed profiles more similar to those in our study; we discussed this paper in the revised manuscript.

2) Lp+ macrophages were described in the study to express a transcriptomic program characterized by M2-like gene program involved in wound repair and healing. Thus, beyond investigating the phagocytic capacity of the macrophage using labeled *E. coli*, can the authors test the uptake of apoptotic cells using an efferocytosis assay, which seems more relevant for wound repair.

We appreciate the Reviewer’s excellent suggestion. We carried out efferocytosis experiments, as advised, included the results in supplementary Figures S6A and B, and describe them in the Results and Methods sections. As shown, the efferocytosis capacity was less prominent than the phagocytosis capacity, but LYVE1-/MHCII^lo^ macrophages were still more capable of efferocytosis than other subgroups.

3) Pg7 "We found that MHCII mRNAs (encoding H2-Ab1, H2-Eb1) divided SKM macrophages into two groups, MHCII-high (Hh) and MHCII-low (Hl) in single-cell profiling analysis." Please provide a figure reference.

The Reviewer makes another great point. We have included UMAPs for H2-Eb1 and H2-Ab1 mRNAs in Figure 3—figure supplement 1A and B.

4) Pg 8, in the immunostaining are different subsets of macrophages localized to different sections of muscle tissues? i.e. more associated with endothelial cells, muscle cells or innervated areas?

We appreciate this important question and have addressed it by performing additional experiments and by demarking the muscle cell borders. First, we indicated the muscle fiber outlines in the revised Figure 4B; this addition helped to identify those macrophages closely located with muscle cells, but only in intermuscular regions, that is, the endomysium and perimysium areas. Second, additional immunofluorescence analysis revealed that the two LYVE1+ macrophage subgroups, LYVE1+/MHCII^lo^ and LYVE1+/MHCII^hi^, but not LYVE1-/MHCII^hi^, were localized close to blood vessels, similar to previous findings in lung interstitial macrophages (Chakarov et al., 2019; Figure 4C, top). Third, the macrophage subgroups LYVE1+/MHCII^lo^, LYVE1+/MHCII^hi^, and LYVE1-/MHCII^hi^ were detected around nerve fibers (Figure 4C, bottom). These findings are described in the Results section.

5) Can the authors provide more insight into both old and young skeletal muscle to determine if an absolute number of macrophages change during aging (for example via flow cytometry) and also to determine using transcriptomics if the subpopulations of macrophages in the study are resident vs non-resident macrophages.

We thank the Reviewer for these questions. The absolute number of live macrophages from SKM for flow cytometry analysis were comparable across young and old, male and female mice. We have included these data in Figure 6—figure supplement 1A–D. Old male mice showed a tendency to have slightly higher levels of macrophages compared to young mice, but the differences were not statistically significant (Figure 6—figure supplement 1A). We described this finding in the Results section. We were unable to unequivocally distinguish resident macrophages from non-resident macrophages, but we believe that our macrophages, selected using markers CD11b and F4/80, are mostly resident macrophages.

6) The study lacks analysis of skeletal muscle macrophages in female young and old mice, and would be more informative if they were included in the study to determine any sex differences. Perhaps qPCR or flow cytometry analysis of the four major subsets can be investigated in female mice, since performing single cell would be cost prohibitive.

We appreciate these helpful comments and suggestions. To address them, we carried out flow cytometric analysis of macrophages from female SKM. We found that female mice also have 4 macrophage subgroups, although the relative ratios were modestly different. We found a statistically significant decrease of SKM macrophage numbers in the LYVE1+/MHCII^hi^ subgroup, and an increase in the LYVE1-/MHCII^lo^ subgroup in female compared to male. We included these data (Figure 4—figure supplement 1A,B), and we explain that we have undertaken dedicated studies to understand the biological significance of the differences in macrophages from male vs female SKM.

7) Pg 11. "Cdk1 and Top2a mRNAs were expressed in an even lower number of macrophages (data not shown)." I am confused by data not shown since the data appears to be in table. If not please show.

We appreciate the careful reading by this Reviewer and apologize for the mistaken description. We have revised the text and indicate that the data are included in the revised Supplementary File 1, Cluster 9.

Reviewer #3 (Recommendations for the authors):1) Manuscript provides no functional insights of how changes in macrophages affect muscle-physiology/pathology in young and aged mice. Conclusion about functions are based on associations.

We appreciate the Reviewer’s remarks. We agree that our characterization of macrophage subpopulations in resting skeletal muscle (SKM) and the changes in subpopulations with age are primarily descriptive. In the revised manuscript, we have toned down our interpretation of the results and have acknowledged that our conclusions are based on associations. In addition, given that we identified new SKM subpopulations and found that their relative abundance changes with age, we propose to expand on these findings. To investigate in depth the functional roles of these macrophage subgroups, we are developing specific genetic mouse models and setting up a number of necessary methodologies. The detailed analysis of these macrophage subgroups represents a long-term effort in our laboratory.

2) Please clarify what the percentage of macrophages in muscle in comparison to other CD45+ cells is.

We thank the Reviewer for this request. In the revised manuscript, Figure 6—figure supplement 1 includes the absolute number of CD45+, CD45+/CD11b+, and CD45+/CD11b+/F4/80+ cells, and the percentage of CD45+/CD11b+/F4/80+ cells in CD45+ cells in young and old mice, males and females, as quantified by flow cytometry.

3) Please clarify wheter the analyzed myeloid cells are present in vessels or in muscle parenchyma. How that affects muscle function is also unclear.

We thank the Reviewer for this important question. To address it experimentally, we have carried out additional immunofluorescence staining and have added more information in Figure 4. First, we have highlighted the muscle membranes in the revised Figure 4B to help visualize the location of macrophages, all of which were found in the intermuscular regions (i.e., endomysium or perimysium areas), and not in the muscle parenchyma. Second, additional immunofluorescence staining revealed that two LYVE1+ macrophage subgroups, LYVE1+/MHCII^lo^ and LYVE1+/MHCII^hi^ (but not LYVE1-/MHCII^hi^), localized close to blood vessels, similar to the previous findings by Chakarov et al., (2019) for lung interstitial macrophages (Figure 4C, top). Third, all 3 detectable macrophage subgroups, LYVE1+/MHCII^lo^, LYVE1+/MHCII^hi^, and LYVE1-/MHCII^hi^ were found around nerve fibers (Figure 4C, bottom). We describe these findings in the Results section. As our studies move forward, we will explore the function of these subgroups and their impact on SKM physiology using appropriate methodologies and genetic mouse models.

4) While the scRNAseq of skeletal muscle macrophages reveals interesting findings about their diversity and how aging affects their transcriptomic profile, the authors chose an unusual approach to analyze the data by defining subsets based on extracellular marker expression (Lyve1 and MHCII) and not transcriptomic profile. By starting with a supervised approach, the authors have missed an important part of their data regarding the high diversity of skeletal muscle macrophages, that cannot only be described through MHC-II and Lyve1 expression. The expression of extracellular markers does not necessarily allow to define functionally distinct subsets. A deeper and more detailed unsupervised analysis is required for scRNA seq data. Moreover, as a starting point, the clusters should be defined based on gene expression and then propose candidate markers to characterize the subpopulations and their functional properties

The Reviewer makes excellent suggestions. We have incorporated his/her helpful advice in the revised manuscript by reanalyzing our scRNA-seq data with updated software, and we present first the unsupervised clustering followed by supervised classification. The unsupervised clustering identified 11 clusters; GO annotation found several clusters that were more reparative, proinflammatory, and phagocytotic, and other clusters that were more proliferative and related to senescence. The unsupervised clusters and featured genes are shown in the revised Figure 1, Table 1, and Supplementary File 1, and are described in the revised text. We then complemented the unsupervised clustering with supervised classification using traditional membrane markers that have been employed to functionally classify macrophages for many years. We agree that the revised approach (first unsupervised classification, and afterwards supervised classification) offers more logical, unbiased, and comprehensive information about SKM macrophages.

5) Several conclusions are not supported by the data – e.g.a) Naming Lyve1+ and Lyve1- macrophages "healing" and pro-inflammatory" is premature as this has not been functionally tested.

We fully agree with the Reviewer and have toned down our conclusions across the manuscript, as advised. For example, we focus on the mRNAs expressed in each macrophage subgroup and simply mention the functions suggested by the corresponding GO annotations.

b) Unless the authors have data showing the general health of the animals used for the scRNAseq, the low number of differentially expressed genes between young and old shouldn't be interpreted as a result of a "healthier" status of the aging cohort. It is rather a common caveat of scRNAseq that lacks sequencing depth. A bulk RNAseq of specific sorted population that changes in aging may be important.

The Reviewer’s advice is spot on. We have modified our description of the scRNA-seq analysis to acknowledge that this type of analysis is shallow and may miss differences that require deeper RNA-seq analysis. Bulk RNA-seq analysis of specific subpopulations is a key component of our future plan, after we have optimized our protocol to harvest enough macrophages of each subgroup, and we have the necessary genetic mouse models in place.

6) Characterization of macrophages with polarization markers is inadequate and based on old literature. Hence, not very informative.

We fully agree with the Reviewer. Following the collective advice from Reviewers and Editors, we started the revised manuscript with unsupervised clustering data yielding 11 clusters of SKM macrophages (Figure 1, Table 1, and Supplementary File 1) To complement this classification, we undertook supervised classification by first assessing traditional markers like MRC1 (CD206) and CD86, but found that more recently identified markers LYVE1 and MHCII (Dick et al., 2022, Chakarov et al., 2019, Lim et al., 2018) were more informative in SKM macrophages. LYVE1 in particular divided SKM macrophages into two similarly sized groups (LYVE1+ and LYVE1-), which showed distinct functions. Therefore, we used LYVE1, along with the common membrane marker MHCII, for supervised classification to complement the unsupervised clustering.

7) The unsupervised analysis of the scRNAseq lacks depth. Each cluster should be described independently of whether they belong to LpHl, LpHh, LnHh and LnHl. This analysis should come as the first figure. For example, do not dismiss the possibility of non-macrophage clusters as cells were only sorted on CD11b marker. Cluster 6 could be a granulocyte cluster.

We appreciate the helpful suggestions from Reviewer 3. We have included more depth in our unsupervised clustering analysis and have moved these data to the beginning of the manuscript (Figure 1, Table 1, and Supplementary File 1), and we later discuss the complementarity of unsupervised and supervised analyses. Regarding the Reviewer’s point on the CD11b marker, we apologize for not explaining this point clearly. We isolated CD11b+ cells by FACS, and performed scRNA-seq analysis on them. But when analyzing the sequencing data, we sought an additional marker, F4/80, in order to be more confident about their identity as macrophages. Thus, we restricted our consideration of macrophages to only those CD11b+/F4/80+ double-positive cells and performed downstream statistical analysis on this group. We addressed the possible contamination by granulocytes in the revised Figure 1—figure supplement 1A; we found that Ly6G+ neutrophils and SiglecF+ eosinophils comprised less than 1% of the total CD11b+/F4/80+ double-positive population. Therefore, we believe that, overall, granulocyte contamination in our macrophage population is very minor.

8) Overall, the logic behind the direction taken for the analysis of the data is unclear:– It is not clear why the authors chose to perform a scRNAseq. If the goals of the authors were to study macrophages subsets based on extracellular markers, sorting these subsets, and performing a bulk RNAseq would have been more adequate and would have provided and better sequencing depth.

We appreciate the Reviewer’s question. Given that the polarization of macrophages is largely tissue-dependent, and that the polarization of SKM macrophages with aging is poorly understood, our primary goals were to identify macrophage subpopulations and investigate how they changed with aging. Given the small number of these macrophages, scRNA-seq analysis has been the best way to answer this question at the moment. To follow better logic, as advised by the Reviewer, we displayed the unsupervised clustering first (Figure 1, Table 1, and Supplementary File 1) and complemented it with supervised analysis afterwards (Figures 2-4). We identified more than 10 subpopulations by unsupervised clustering and supervised classification, many of which were not previously reported. We further observed altered genes during aging, and identified specific subpopulations, particularly Cl6 and Cl8 clusters, and LYVE1+ and LYVE1- subgroups in old SKM.

As Reviewer 3 correctly points out, bulk RNA-seq analysis with sorted macrophage subgroups will be informative, especially because the sensitivity of scRNA-seq is low. We plan to expand this analysis as our studies progress and we are able to collect sufficient numbers of macrophages and have appropriate mouse models in hand to address these questions.

– The choice of MHC-II and Lyve1 markers to divide macrophage subset seems arbitrary. It is not clear why the authors chose to use markers described for lung macrophages specifically.

We appreciate this question. We investigated LYVE1 and MHCII because they effectively classified many tissue-resident macrophages in recent published reports – not only in lung, but also in artery, heart, brain, kidney, and liver (Lim et al., 2018, Chakarov et al., 2019, Dick et al., 2022). Empirically, we observed that LYVE1 (but not other markers examined) divided SKM macrophages into two similarly sized groups that showed distinct functions in association analyses in SKM. Traditional polarization markers, e.g., MRC1 (CD206) and CD86 were expressed in the vast majority of our SKM macrophages, and most macrophages were positive for both as shown in Figure 1—figure supplement 1C [similar expression patterns were shown by Kosmac et al. (2018) in human SKM macrophages]. These observations indicated that traditional polarization markers may not be suitable for classification of SKM macrophages specifically. Therefore, for SKM macrophages, we used LYVE1 and MHCII as membrane polarization markers.

– The data showing phagocytic capacities are interesting. Though, it is not clear why the authors chose to investigate this function. Is phagocytosis relevant to the physiological function of skeletal muscle tissues? Are there other functions highlighted in the GO annotations that can be tested? (endocytosis, inflammatory response, antigen presentation)

The Reviewer brings up important questions. After the scRNA-seq analysis revealed an interesting heterogeneity and polarization status of SKM macrophages, we sought to study the biological function of the macrophage subgroups. We focused on phagocytosis because it is a key function of macrophages during the repair and defense of injured SKM. However, prompted by the advice of Reviewer 2 and questions from Reviewer 3, we carried out experiments that focused on efferocytosis (the capacity of macrophages to engulf apoptotic cells). Efferocytotic capacities were generally less robust than phagocytic capacities (Figure 5—figure supplement 1A,B), but here too, efferocytosis was more vigorous in the LYVE1-/MHCII^lo^ subgroup than the other 3 subgroups, similar to the findings with phagocytosis. We included the efferocytosis data in Figure 5—figure supplement.

9) It is difficult to evaluate the solidity of the data. For quantification of populations by flow cytometry analysis and using the scRNAseq data, we suggest adding histograms representing n numbers and statistical significance when comparing the abundance of populations (Figure 3D, S3C, 5A, 6A). Moreover, instead of using tables to depict differentially expressed genes, heatmaps or volcanoplots are recommended (Figure 5C, S3C).

We appreciate the Referee’s helpful advice. We agree that the suggested representations are far more intuitive. In the revised manuscript, we display the data using histograms, heatmaps, UMAPs, violin plots, etc., as shown in the revised Figures 1E, 4A, 6C, 7A-D, and Figure 1—figure supplement 2B,C, Figure 5—figure supplements 1 and 2, Figure 6—figure supplement 1, Figure 7—figure supplements 1 and 2.

Figure 1C and Figure 2: Ln cells appear to be multiple clusters – with same features? How is that possible?

We thank the Reviewer for his/her keen eye. Unsupervised clustering revealed that LYVE- cells (Ln by the old nomenclature) comprised ~7 smaller clusters, Cl3-Cl9 (Table 1). By contrast, unsupervised clustering of LYVE1+ yielded only two subgroups (Cl0 and Cl1). As discussed in the Results section, these findings suggest that LYVE1- (M1-like) macrophages appear to be more diverse than LYVE1+ (M2-like) macrophages. In keeping with this notion, M2 markers were quite robust (e.g., LYVE1, MRC1, CD163 etc.), while M1 markers were generally less useful. In this regard, providing gene expression details of the unsupervised and supervised subgroups helps readers understand the heterogeneity of SKM macrophages.

Figure 1D: The bioinformatic analyses used to characterize Lp/Ln cells lacks statistical validity.

We thank the reviewer for this comment. We removed the old Figure 1D, and merged those data with data that are now displayed in the revised Figures 2B,C. We also listed featured genes in Supplementary File 2, so that future readers that can easily access the most current version of g:Profiler to find GO annotations.

Figure 3D: biological purpose to reveal new insights is lacking.

We appreciate the request to elaborate on the biological purpose (note: the original Figure 3D is now Figure 4A). Identifying two new subgroups, LYVE1-/MHCII^lo^ and LYVE1+/MHCII^hi^, provided important insights that we plan to expand into future studies. The LYVE1-/MHCII^lo^ subgroup formed a separate cell cluster, as shown in the new Figure 4A, and GO annotation further suggested a cell killing function of this subgroup (Figure 3C); interestingly, the size of this subgroup increased in old SKM (Figure 7A,B). The possible involvement of this subgroup in cleaning up senescent cells in aging SKM is an exciting future direction for our team.

The LYVE1+/MHCII^hi^ subgroup showed an ‘intermediate’ status by flow cytometric analysis. This subgroup may change phenotypes depending on environmental cues and is another important subgroup that we will investigate as our studies advance. In sum, our study offers important insight into the function of macrophage subpopulations in SKM aging.

Figure 3E; Fails to provide spatial clarity of localization in muscle.

In the revised Figure 4B, we marked the contour of muscle cells, so that readers can recognize more easily that the macrophages appear to be located in the intermuscular perimysium and endomysium areas. In Figure 4C, we have included additional immunofluorescence analysis showing that LYVE1+ macrophages are closely localized with blood vessels, while both LYVE1+ and LYVE1- macrophages are located near nerve fibers.

Figure 4: phagocytic analyses are superficial and fail to provide new insights.

Thank you for the opportunity to expand on the phagocytic analysis and discuss the results. We anticipated seeing strong phagocytic activity across the macrophage subgroups, but it was unexpected to find the particularly strong phagocytotic capacity of LYVE1-/MHCII^lo^ macrophages. The same subgroup also showed stronger efferocytotic capacity in an additional experiment that was carried out during the revision (Figure 5—figure supplement 1). These results offer strong evidence that the function of LYVE1-/MHCII^lo^ macrophages could include ‘cleaning-up’ during SKM aging, an exciting trait that we plan to pursue with appropriate mouse models.

Figure 6: is the most important figure. However, it is poorly put together in terms of data analyses and presentation. Identity of cluster 5 and 6 which change with age is the main finding – but their relevance is unknown

We appreciate the Reviewer’s comment. Most of the content in the original Figure 6 is now included in Figure 7. The new Figure 7 highlights the changes in macrophage numbers and expression levels of key genes in clusters during aging. As the Reviewer notes, changes in Cl5 and Cl6 (now Cl6 and Cl8) during aging were more pronounced. Many senescence-associated mRNAs including Gpnmb, Spp1, Ctsd mRNAs were found in the new Cl6, and S100a8 and S100a9 mRNAs, encoding non-classical proinflammatory markers, were enriched in the new Cl8. The elevated expression of these and other mRNAs in these two clusters in old SKM suggests that they are involved in SKM aging. As the Reviewer points out, their relevance is not known at present, and further studies are needed to characterize them fully.

[Editors’ note: further revisions were suggested prior to acceptance, as described below.]

Upon discussion, the reviewers thought that this could be addressed with:1. Doing an analysis paralleling the main manuscript but using only the 2 samples that were processed "unbatched" (i.e. the samples where 1 young and 1 old mouse were processed in parallel), and only these. If the main results of the study are conserved in this unbatched subset of the data, this would strengthen the likelihood that the batching did not grossly impact the conclusions. We would then recommend including this analysis as a supplement.

We sincerely appreciate the helpful guidance from Reviewers and Editors. We shared their concern about possible batch effects, as the lengthy protocol for isolating macrophages from mouse skeletal muscle allowed us to process only one mouse at a time, and two mice total in one day by the same persons. Following the requested additional analyses, we gained evidence to support the notion that batch effects did not grossly impact the conclusions. We have provided the following additional analyses in this revision:

1. In the new Figure 1—figure supplement 1B, we now show the distribution of macrophages in the 11 clusters across all replicates individually (Y1, Y2, Y3, O1, O2, O3) and the combined replicates (Y1-3, O1-3) both young and old. We hope these side-by-side comparisons help the readers appreciate the similarities between batches/replicates.

2. In the new Figure 1—figure supplement 1C, we have included UMAP analyses of all biological replicates for both young and old. Again, these analyses can help the readers evaluate the similarities in gene expression patterns across batches/replicates.

3. In the new Figure 3—figure supplement 1B, we now display separately the biological replicates of macrophage distribution in four subgroups following a supervised classification. We show each individual sample (Y1, Y2, Y3, O1, O2, O3) and the combined samples (Y1-3, O1-3). In Figure 3—figure supplements 1B and 1C, the readers can see side-by-side the replicates in the supervised classification analysis, for both cell distribution and gene expression patterns.

4. In the revised Materials and methods section, we explicitly indicate that the isolated SKM macrophages were immediately subjected to single-cell library construction without culture or treatment. Given the methodological hurdles with the lengthy protocol, we believe that this strategy helped to maintain the in vivo characteristics and reduce batch-related problems.

It is worth noting the consistency in expression of other specific genes across replicates in young and old macrophages (Figure 7—figure supplement 2). Similarly, flow cytometric analysis using antibodies against LYVE1 and MHCII showed patterns very similar to those observed by the supervised classification of single-cell transcriptomics (percentages in Figure 7A,B). In sum, any potential batch effects by the collection scheme did not influence the results, and the main conclusions remain unchanged.

2. As highlighted by reviewer #3, it is crucial that the batching/experimental collection scheme be discussed explicitly in the manuscript.

We fully agree with this important request. In the revised Results, when describing Figures 1 and 3 (pages 5 and 9), we have discussed the additional analysis carried out to alleviate concerns related to batch effects.

3. Finally, please address Reviewer #1's remaining concern on the use of a background gene list for the g:profiler analysis.

We appreciate the request for additional analysis requested by Reviewer #1 and have addressed it by using genes detected from our SKM macrophages as background gene set, as requested. Our response to the reviewer is as follows:

“We appreciate the Reviewer’s following up on this request and apologize for previously misunderstanding this question, as indeed we had mistakenly used all mouse genes as the background in our earlier analysis. Following the Reviewer’s suggestion, we now used instead the 14,542 genes detected in our young and old skeletal muscle macrophages (CD11b+/F4/80+) as the background gene set. We included this information in the revised Materials and methods section (page 20). We ran again all GO annotations for this revision and obtained slightly different functional terms with the new background gene set. For example, we found that only SubCl0 in the LYVE1^-^/MHCII^lo^ subgroup shows phagocytosis-related terms in the new analysis (Figure 5—figure supplement 2). We included the new analysis as a table in Figure 5—figure supplement 2, and we removed the Supplementary File 4 we had included in the previous submission. We also updated Figures 2B, 2C, 3C, and 6B, as well as Table 1, the Results section, and Figure Legends section.”

Reviewer #1 (Recommendations for the authors):Although the authors have addressed most of my concerns, some large concerns remain at this point.1. A very large concern was revealed by their answer to one of my questions about the batchiness of the data. Indeed, the author's response revealed 3 batched: (i) only young samples, (ii) only old samples and (ii) one old and one young. Unfortunately, since batch and biological groups are confounded for groups i and ii, that data is meaningless (i.e. batch cannot be properly accounted for when it is confounded with biology). Since I understand that the authors may not be able to redo the entire experiment the way it should have been done, I believe it is imperative that all analyses also be done exclusively on batch 3 (the one where both groups were represented), to show that all results would hold in the absence of batch. The results should then be included and compared/discussed in the context of the paper as this is a big problem.

We appreciate the helpful guidance from the Reviewer. We shared his/her concern about possible batch effects, stemming from the fact that the lengthy isolation of macrophages from mouse skeletal muscle allowed us to process only one mouse at a time, and two mice total in one day by the same persons. By performing the requested additional analyses, we gained evidence supporting the notion that batch effects did not grossly impact the conclusions. We have provided the following additional analyses in this revision:

1. In the new Figure 1—figure supplement 1B, we now show the distribution of macrophages in the 11 clusters across all replicates individually (Y1, Y2, Y3, O1, O2, O3) and the combined replicates (Y1-3, O1-3) both young and old. We hope these side-by-side comparisons help the readers appreciate the similarities between batches/replicates.

2. In the new Figure 1—figure supplement 1C, we have included UMAP analyses of all biological replicates for both young and old. Again, these analyses can help the readers evaluate the similarities in gene expression patterns across batches/replicates.

3. In the new Figure 3—figure supplement 1B, we now display separately the biological replicates of macrophage distribution in four subgroups following a supervised classification. We show each individual sample (Y1, Y2, Y3, O1, O2, O3) and the combined samples (Y1-3, O1-3). In Figure 3—figure supplements 1B and 1C, the readers can see side-by-side the replicates in the supervised classification analysis, for both cell distribution and gene expression patterns.

4. In the revised Materials and methods section, we explicitly indicate that the isolated SKM macrophages were immediately subjected to single-cell library construction without culture or treatment. Given the methodological hurdles with the lengthy protocol, we believe that this strategy helped to maintain the in vivo characteristics and reduce batch-related problems.

It is worth noting the consistency in expression of other specific genes across replicates in young and old macrophages (Figure 7—figure supplement 2). Similarly, flow cytometric analysis using antibodies against LYVE1 and MHCII showed patterns very similar to those observed by the supervised classification of single-cell transcriptomics (percentages in Figure 7A,B). In sum, any potential batch effects by the collection scheme did not influence the results, and the main conclusions remain unchanged.

2. The authors still did not address the background list used for GO enrichment in g:profiler. This leads me to believe they used the default (all genes instead of detected genes in the dataset), which is incorrect and would lead to spurious enrichments. These analyses should be rerun with the correct background list.

As mentioned above in our response to the Editors, we appreciate the Reviewer’s following up on this request and apologize for previously misunderstanding this question, as indeed we had mistakenly used all mouse genes as the background in our earlier analysis. Following the Reviewer’s suggestion, we now used instead the 14,542 genes detected in our young and old skeletal muscle macrophages (CD11b+/F4/80+) as the background gene set. We included this information in the revised Materials and methods section (page 20). We ran again all GO annotations for this revision and obtained slightly different functional terms with the new background gene set. For example, we found that only SubCl0 in the LYVE1^-^/MHCII^lo^ subgroup shows phagocytosis-related terms in the new analysis (Figure 5—figure supplement 2). We included the new analysis as a table in Figure 5—figure supplement 2, and we removed the Supplementary File 4 we had included in the previous submission. We also updated Figures 2B, 2C, 3C, and 6B, as well as Table 1, the Results section, and Figure Legends section.

Reviewer #2 (Recommendations for the authors):We believe the authors have responded to the concerns of the reviewers sufficiently and the paper is significantly improved. Thus, in our opinion, the paper is suitable for publication.

We appreciate the positive comments from Reviewer #2.

Reviewer #3 (Recommendations for the authors):The authors have addressed most of my prior concerns. Some issues remain, but in general given the importance of the topic, the mansucript is ready to forward in the process.An important issue that remains unaddressed is that the scRNA analyses and cell sorting for young/old groups were done on different days. The authors responded to this issue and acknowledged this caveat, but do not describe the consequence of this on the data generation and conclusions.Authors need to provide this information. This reviewer could not find it in the beginning of the Results section where the description of data generation is provided.

We appreciate the helpful comments from the Reviewer. Following the advice and suggestions from the Editors and Reviewers, we have incorporated analysis that explicitly addresses if possible batch effects modify the overall conclusions. As we explain above in our responses to the Editor and Reviewer 1, the lengthy protocol for isolating macrophages from mouse skeletal muscle made it impossible for us to process more than one mouse at a time, and only two mice total could be processed per day by the same persons. Therefore, we have done the requested additional analyses to ensure that such effects did not grossly impact the conclusions. We have provided the following additional analyses in this revision:

1. In the new Figure 1—figure supplement 1B, we now show the distribution of macrophages in the 11 clusters across all replicates individually (Y1, Y2, Y3, O1, O2, O3) and the combined replicates (Y1-3, O1-3) both young and old. We hope these side-by-side comparisons help the readers appreciate the similarities between batches/replicates.

2. In the new Figure 1—figure supplement 1C, we have included UMAP analyses of all biological replicates for both young and old. Again, these analyses can help the readers evaluate the similarities in gene expression patterns across batches/replicates.

3. In the new Figure 3—figure supplement 1B, we now display separately the biological replicates of macrophage distribution in four subgroups following a supervised classification. We show each individual sample (Y1, Y2, Y3, O1, O2, O3) and the combined samples (Y1-3, O1-3). In Figure 3—figure supplements 1B and 1C, the readers can see side-by-side the replicates in the supervised classification analysis, for both cell distribution and gene expression patterns.

4. In the revised Materials and methods section, we explicitly indicate that the isolated SKM macrophages were immediately subjected to single-cell library construction without culture or treatment. Given the methodological hurdles with the lengthy protocol, we believe that this strategy helped to maintain the in vivo characteristics and reduce batch-related problems.

It is worth noting the consistency in expression of other specific genes across replicates in young and old macrophages (Figure 7—figure supplement 2). Similarly, flow cytometric analysis using antibodies against LYVE1 and MHCII showed patterns very similar to those observed by the supervised classification of single-cell transcriptomics (percentages in Figure 7A,B). In sum, any potential batch effects by the collection scheme did not influence the results, and the main conclusions remain unchanged.